# Super-resolution imaging in whole cells and tissues via DNA-PAINT on a spinning disk confocal with optical photon reassignment

Cecilia Zaza [1], Megan D. Joseph[1], Olivia P. L. Dalby [1,2], Rhian F. Walther[3], Karol Kołątaj [4,5], Germán Chiarelli[4], Franck Pichaud[3,6], Guillermo P. Acuna [4,5] & Sabrina Simoncelli [1,2]

Single-Molecule Localization Microscopy (SMLM) has traditionally faced challenges to optimize signal-to-noise ratio, penetration depth, field-of-view (FOV), and spatial resolution simultaneously. Here, we show that DNA-PAINT imaging on a Spinning Disk Confocal with Optical Photon Reassignment (SDC-OPR) system overcomes these trade-offs, enabling high-resolution imaging across multiple cellular layers and large FOVs. We demonstrate the system's capability with DNA origami constructs and biological samples, including nuclear pore complexes, mitochondria, and microtubules, achieving a spatial resolution of 6 nm in the basal plane and sub-10 nm localization precision at depths of 9 μm within a $53 \times 53$ μm² FOV. Additionally, imaging of the developing *Drosophila* eye epithelium at depths up to 9 μm with sub-13 nm average localization precision, reveals distinct E-cadherin populations in adherens junctions. Quantitative analysis of Collagen IV deposition in this epithelium indicated an average of $46 \pm 27$ molecules per secretory vesicle. These results underscore the versatility of DNA-PAINT on an SDC-OPR for advancing super-resolution imaging in complex biological systems.

Single-molecule localization microscopy (SMLM)[1] techniques have significantly advanced our understanding of fundamental biological processes by providing exceptional resolution. Although powerful, the implementation of SMLM has often been limited to selective illumination configurations to meet the high signal-to-noise ratio (SNR) requirements for single-molecule detection, thus imposing a compromise between penetration depth, field-of-view (FOV), and spatial resolution. The most widely used implementation of SMLM is in combination with wide-field illumination, particularly under total internal reflection (TIR)[2] or highly inclined and laminated optical sheet (HILO)[3] excitation. These methods routinely achieve lateral localization precisions ($\sigma_{SMLM}$) below 5 nm[4] using DNA-PAINT imaging[5], and in recent advancements, have demonstrated Ångström-scale resolution when combined with sequential imaging[6]. However, these approaches come at the expense of either limited penetration depth, <250 nm for TIR excitation, or small FOV of ~40 × 40 μm² for HILO excitation[7,8].

Confocal-based configurations, including point-scanning and spinning disk confocal (SDC), have also been successfully paired with SMLM techniques. By physically blocking out-of-focus light with the use of pinholes they provide the advantage of deep sample penetration (up to 100 μm)[9], making them preferable for imaging tissue samples. However, point-scanning configurations are not commonly used for SMLM due to their inherently slow imaging acquisition speed, as image formation is realized by progressively scanning single focuses over a sample. This results in relatively small FOVs of 20 × 20 μm² and can reach resolutions of 20 nm[10,11]. The integration of an array detector

[1]London Centre for Nanotechnology, University College London, London, UK. [2]Department of Chemistry, University College London, London, UK. [3]Laboratory for Molecular Cell Biology, University College London, London, UK. [4]Department of Physics, University of Fribourg, Fribourg, Switzerland. [5]Swiss National Center for Competence in Research (NCCR) Bio-inspired Materials, University of Fribourg, Fribourg, Switzerland. [6]Institute for the Physics of Living Systems, University College London, London, UK. ✉e-mail: s.simoncelli@ucl.ac.uk

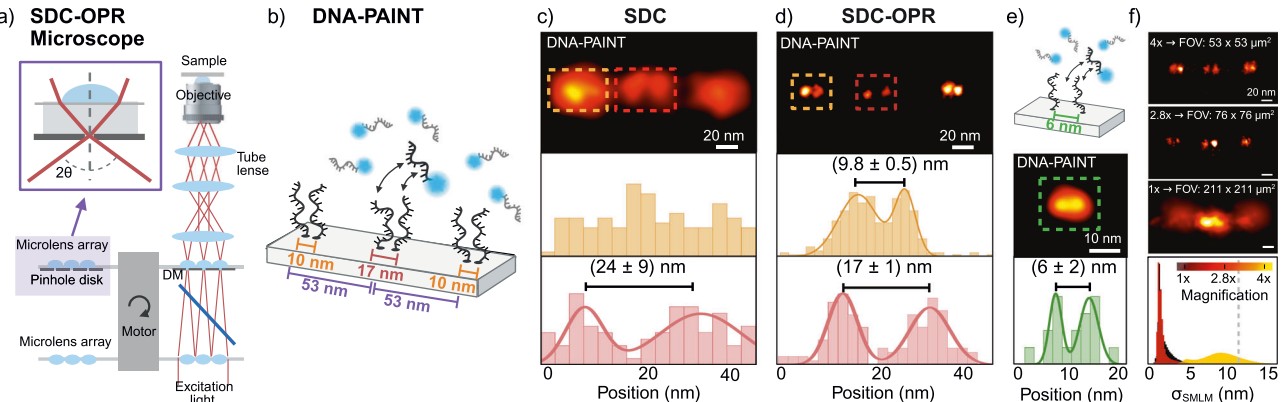

**Fig. 1 | DNA-PAINT imaging of DNA origami structures on an SDC-OPR microscope. a** Schematic of the Spinning Disk Confocal with Optical Photon Reassignment (SDC-OPR) system (CSU-W1 SoRA Nikon), featuring a microlens array designed to minimize pinhole size and maximize photon collection. **b** DNA origami design with six DNA-PAINT docking strands arranged in three pairs: 53 nm apart overall; central pair exhibiting the largest separation of 17 nm, while the edge pairs are spaced 10 nm apart. Detailed sample preparation and imaging conditions are provided in the Methods section and Supplementary Table 1. Staple modifications can be found in Supplementary Table 2. **c** Spinning Disk Confocal (SDC) DNA-PAINT image of a representative DNA origami (top), and position distribution for the left (top) and central (bottom) docking strands pairs, showing measured distances. **d** SDC-OPR DNA-PAINT image of a representative DNA origami (top), and position distribution of the left (top) and central (bottom) docking pairs,

showing measured distances. **e** DNA origami design featuring 6 nm-spaced docking sites to assess achievable resolution (top). Staple modifications can be found in Supplementary Table 3. DNA-PAINT image of a representative DNA origami (central), and position distribution for the docking pair, highlighting measured distance (bottom). **f** Localization precision ($\sigma_{SMLM}$) across three different FOVs sizes enabled by the SDC-OPR system: 53 μm, 76 μm and 211 μm side size. Representative super-resolved DNA origami images corresponding to each FOV size are shown in the three top panels. The bottom panel displays the distribution of localization precision ($\sigma_{SMLM}$) as histograms, with different colors for each magnification. All DNA origami sample imaging were repeated in two independent experiments. Source data of histograms is provided as a Source Data file. Panel (a) and DNA origami sketches of panel (b) and (e) were created with BioRender.com.

in confocal-based configurations enables to perform Image Scanning Microscopy (ISM)[12], which doubles the spatial resolution of conventional confocal microscopy via pixel reassignment. Recently, the combination of ISM with SMLM has achieved localization precision of 6 nm in the basal plane. However, this comes with the limitation of a relatively small FOV of $8 \times 8\ \mu m^2$ [13].

To increase acquisition speed and FOV size, spinning disk confocal (SDC) systems employ hundreds of spiraled pinholes on a rotating opaque disk, coupled with a camera instead of a single-point detector. Recent implementations of SDC configurations with SMLM have achieved planar localization precision as high as 8 nm for DNA origami samples and 22 nm for cell samples up to 5 μm in depth using DNA-PAINT[14]. However, as the emission light passes through the disks, which enhances optical sectioning by rejecting out-of-focus fluorescence, this also reduces photon collection, ultimately constraining the achievable resolution. Light sheet illumination offers an alternative approach for deep imaging with high SNR across large FOVs. Yet, its use of low numerical aperture (NA: 1.0) objectives in dual-objective setups results in reduced localization precision. Current applications of light sheet illumination in combination with dSTORM or DNA-PAINT in whole cells or thin tissue samples report localization precisions that are approximately five times worse than those achieved with TIRF, around 20 nm—comparable to the resolution obtained with SDC systems[15-17].

In 2015, Azuma and colleagues proposed an enhanced SDC with optical photon reassignment (SDC-OPR)[18]. Unlike computational or hardware-based pixel reassignment, this approach involves adding a set of microlenses to the disk of the original SDC configuration, as depicted in Fig. 1a. The microlenses contract the focus two-fold while maintaining the orientation of the focus[19]. This focus contraction redirects emitted photons to their most probable points of origin, thereby improving overall photon collection. Therefore, this raises the question as to whether SCD-OPR can outperform current optical configurations, overcoming the trade-offs between penetration depth, field-of-view, and spatial resolution.

Here, we demonstrate that SMLM performed on a commercial SDC-OPR microscope can achieve sub-2 nm localization precision in the basal plane and sub-10 nm up to 9 μm penetration depth, within a $53 \times 53\ \mu m^2$ FOV. The power of the SDC-OPR system for SMLM imaging is further highlighted by its ability to achieve an average $\sigma_{SMLM}$ of $(12 \pm 1)$ nm when imaging adherens junctions (AJ) in the developing *Drosophila* retinal epithelium over 9 μm depth. This revealed the adhesion molecule E-cadherin is heterogeneously distributed at the AJ, suggesting this adhesion compartment contains different E-cadherin populations.

## Results

### DNA-PAINT on an SDC-OPR achieves 6 nm resolution with DNA origami samples

To evaluate the spatial resolution achievable via SMLM on an SDC-OPR system, we designed 2D DNA origami nanostructures suitable for DNA-PAINT imaging. These structures feature three pairs of DNA docking strands spaced 53 nm apart, with edge pairs separated by 10 nm and the central pair by 17 nm, as illustrated in Fig. 1b. Figure 1c shows an image of a single DNA origami acquired via DNA-PAINT on a standard SDC microscope, achieving an average $\sigma_{SMLM}$ of 10 nm consistent with previous studies[14]. The bottom panel of Fig. 1c shows the histogram of localization position for the center and edge pairs of docking strands (both edge pairs are identical). As can be seen from the distribution of localization positions, with the SDC, we could not resolve individual docking strands spaced 10 nm apart. More representative DNA origami images can be found in Supplementary Fig. 1a. In contrast, DNA-PAINT on the SDC-OPR (Fig. 1d) successfully resolved individual docking strands, both separated by 17 nm and 10 nm, across all DNA origami structures within a $53 \times 53\ \mu m^2$ FOV. A representative single DNA origami imaged on the SDC-OPR system is displayed in Fig. 1d (top panel), with more examples shown in Supplementary Fig. 1b. We measured average distances of $(17 \pm 1)$ nm and $(9.8 \pm 0.5)$ nm for central and edge docking strand pairs, respectively (Fig. 1d bottom panel), consistent with our observations under TIR illumination (Supplementary Fig. 1e and Supplementary Fig. 2). Notably, SDC-OPR achieved an exceptional

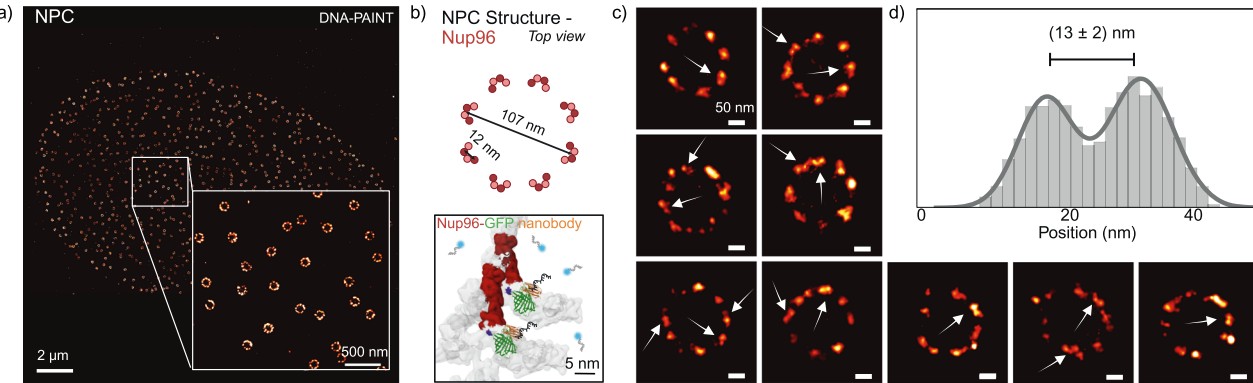

**Fig. 2 | DNA-PAINT imaging of Nup96 in nuclear pore complex on an SDC-OPR microscope. a** Representative DNA-PAINT image of nuclear pore complexes (NPCs) in a U2OS cell, acquired using the SDC-OPR system. The imaging was performed using a DNA-labeled anti-GFP nanobody targeting Nup96, which is fluorescently tagged with mEGFP. Detailed sample preparation and imaging conditions are provided in the Methods section and Supplementary Table 1. A zoomed-in region emphasizes the resolution of individual NPCs. **b** Structural representation of Nup96 proteins (red) within NPCs (gray) adapted from Cryo-EM data (PDB 7PEQ).

This depiction illustrates the GFP labeling of Nup96 and the corresponding DNA-labeled anti-GFP nanobody utilized in the DNA-PAINT imaging of NPCs. **c** Examples of individual NPCs identified in the SDC-OPR DNA-PAINT imaging, with arrows indicating specific pairs of Nup96 proteins. **d** Cross-sectional histogram displaying the average distances between protein pairs ($n = 16$) located within single symmetry centers of NPCs, highlighting the spatial organization of Nup96. Nup96 imaging was repeated in three independent experiments. Source data of histograms are provided as a Source Data file. Panel (b) was created with BioRender.com.

$\sigma_{SMLM}$ of 1.4 nm as calculated by the Cramér-Rao lower bound (CRLB) of the single-molecule fits and 2.3 nm localization precision as calculated by nearest-neighbor-based metric (NeNA)[20]. This high level of localization precision achieved in a confocal-based system via DNA-PAINT imaging on the SDC-OPR, even enabled the resolution of DNA docking strands separated by just 6 nm (Fig. 1e and Supplementary Fig. 3).

The bottom panel of Fig. 1f depicts the distributions of $\sigma_{SMLM}$ obtained for the different magnifications available in the commercial SDC-OPR microscope (CSU-W1 SoRA Nikon system). A similar $\sigma_{SMLM}$ of 1.8 nm and NeNA parameter of 3.1 nm was maintained for $76 \times 76 \, \mu m^2$ FOVs. This result is particularly noteworthy given that these microscopes are typically equipped with low-power lasers to minimize photobleaching in biological samples. Despite nearly doubling the imaging area, the localization precision remained practically unchanged, highlighting the robustness of the system in achieving high-resolution over extended FOVs without compromising image quality. However, as the FOV increased to $211 \times 211 \, \mu m^2$, the localization precision increased slightly to 8 nm due to the reduced excitation power density. Supplementary Fig. 1 provides additional images and quantitative data under various imaging conditions. Figure 1f also shows representative individual DNA origami structures imaged for each of the magnifications for direct comparison. These images highlight that the center and edge pairs can be resolved with magnification 4× ($53 \times 53 \, \mu m^2$) and 2.8× ($76 \times 76 \, \mu m^2$), while at 1× magnification ($211 \times 211 \, \mu m^2$), only the center pair becomes discernible. These results demonstrate superior resolution in confocal-based microscopy, capable of distinguishing structures closer than 10 nm.

### DNA-PAINT on an SDC-OPR resolves 12 nm Nup96 pairs distance in U2OS Cells

To illustrate the enhanced resolution achieved with an SDC-OPR in a cellular context, we imaged the structural proteins of the nuclear pore complexes (NPCs) in U2OS cells. The NPC, a large multi-protein structure, serves as the major gatekeeper of nucleocytoplasmic transport, controlling the movement of molecules between the nucleus and the cytoplasm. NPCs were selected as a model system due to their stereotypic protein arrangement, often used to benchmark super-resolution microscopy[21]. Figure 2a shows a representative DNA-PAINT SDC-OPR image of nucleoporin 96 (Nup96), tagged with monomeric enhanced green fluorescent protein (mEGFPs) and labeled

with DNA-conjugated anti-GFP nanobodies, in U2OS cells. Nup96, a structural protein of the NPC's Y-complex, is present in eight pairs on both the cytoplasmic and nuclear rings, totaling 32 copies per NPC (Fig. 2b, top). We imaged the cytoplasmic ring, where individual Nup96 protein pairs are spaced 12 nm laterally as depicted in Fig. 2b. Zoomed-in on selected NPCs reveals distinct pairs of closely spaced Nup96 proteins (indicated by arrows in Fig. 2c, with more examples presented in Supplementary Fig. 4a). We measured the Euclidean distance between Nup96 pairs by aligning them and plotting the cross-sectional histogram of the summed image ($n = 16$ pairs). This reveals a peak-to-peak distance of $(13 \pm 2)$ nm shown in Fig. 2d, consistent with EM modes and HILO[22] and TIR imaging results (Supplementary Figs. 4b and 5). Additionally, each peak fit exhibits a 4 nm standard deviation confirming the high $\sigma_{SMLM}$ of 3.3 nm achieved via DNA-PAINT on SDC-OPR (NeNA = 4.4 nm). We note that this high level of localization precision was achieved across the whole (FOV) of $53 \times 53 \, \mu m^2$, which is remarkable for confocal-based set-ups.

### Multiplexed and ultra-high-resolution imaging of Nup96 pairs via Exchange-PAINT and RESI on an SDC-OPR system

To further demonstrate the versatility of DNA-PAINT imaging on an SDC-OPR system, we performed multiplexed imaging of alpha-tubulin, mitochondria and Nup96 proteins in U2OS cells using the Exchange-PAINT technique[23]. Exchange-PAINT utilizes orthogonal DNA-imager strands conjugated to a single fluorophore, allowing all targets to be excited by the same laser source. Instead of simultaneous acquisition, signals are separated across sequential imaging rounds, simplifying its integration into any commercial SDC-OPR system. Remarkably, the exceptional single-molecule localization precision achieved for single-color Nup96 imaging was maintained across all targets, with $\sigma_{SMLM}$ values of 3.9 nm, 4.0 nm and 3.3 nm for the nuclear-pore-complex, alpha-tubulin and mitochondria, respectively, in the multi-target experiment (Fig. 3a).

The integration of Exchange-PAINT into an SDC-OPR system also enabled the implementation of Resolution Enhancement by Sequential Imaging (RESI)[6]. RESI combines DNA barcoding with sequential imaging to isolate and group localizations of individual molecular targets. This approach achieves Ångström-level precision in determining the spatial positions of targets, making it particularly effective for resolving protein complexes with separations well below 10 nm. Previously demonstrated under wide-field excitation, we implemented RESI in the SDC-OPR system using Nup96 protein pairs for benchmarking.

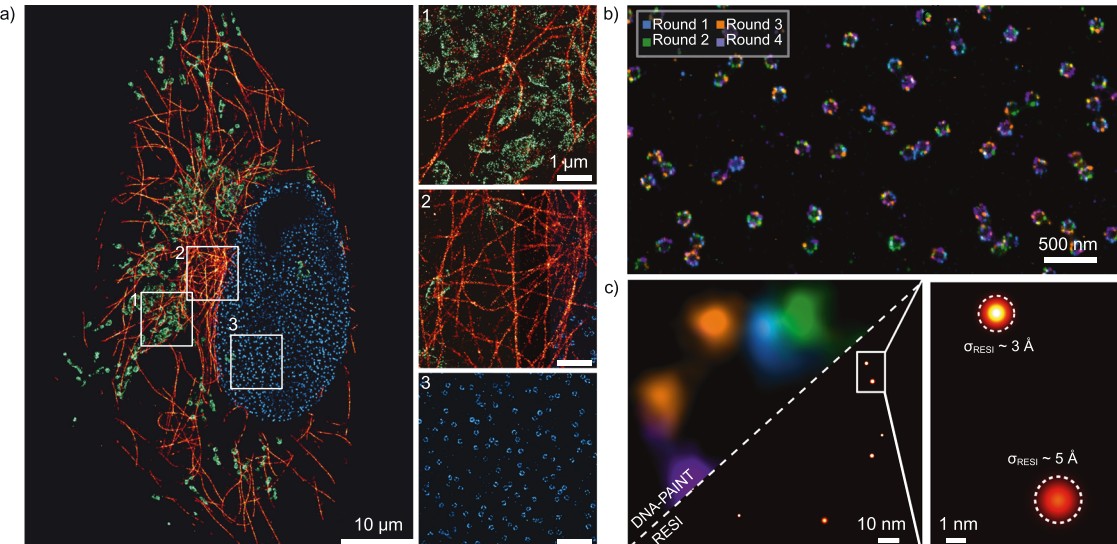

**Fig. 3 | Exchange-PAINT imaging for multicolor and RESI imaging using an SDC-OPR microscope. a** Three-target Exchange-PAINT imaging of microtubules (red), mitochondria (green) and nuclear pore complexes (blue) in U2OS cells, acquired using the SDC-OPR system. Microtubules were imaged via DNA-conjugated anti-alpha-tubulin. NPCs were imaged via Nup96, tagged with mEGFP and labeled with DNA-conjugated anti-GFP nanobodies. Mitochondria were imaged via DNA-conjugated secondary nanobody against TOM20. Sample preparation is described in the methods and imaging conditions are summarized in Supplementary Table 1. Zoomed-in region emphasizes the enhanced resolution obtained on the SDC-OPR system focusing on (1) mitochondria, (2) microtubules and (3) nuclear pore complexes. Scale bars = 1 μm. **b** Exchange-PAINT image of Nup96-mEGFP proteins stochastically labeled with orthogonal DNA sequences by incubation of the sample with anti-GFP nanobodies, each conjugated with one of four orthogonal sequences. Color represents different rounds of imaging. **c** Comparison of DNA-PAINT (left) and Resolution Enhancement by Sequential Imaging (RESI, right) for a single NPC illustrating improvement in spatial resolution by RESI. Localizations are rendered as Gaussians with DNA-PAINT localization precision ($\sigma_{DNA\text{-}PAINT}$) and RESI localization precision ($\sigma_{RESI}$), respectively. A zoomed-in view of a single Nup96 pair, showcasing the enhanced $\sigma_{RESI}$. Multicolor imaging was repeated in two independent experiments.

Following the RESI workflow, we labeled Nup96-mEGFP molecules stochastically with four orthogonal DNA-conjugated anti-GFP nanobodies. Sequential DNA-PAINT imaging over four rounds generated sufficiently spaced localization groups corresponding to individual Nup96 target molecules (Fig. 3b). Super-localization analysis using the RESI algorithm resolved individual Nup96 proteins across the entire FOV (more examples presented in Supplementary Fig. 6). RESI reconstruction achieved an average lateral localization precision of ~3 Å, representing a ten-fold improvement over single-round DNA-PAINT imaging (Fig. 3c). These results highlight RESI's ability to achieve label-size-limited resolution within an SDC-OPR framework.

### SDC-OPR provides high-resolution DNA-PAINT imaging across large fields-of-view and cell depths

After demonstrating the exceptional resolution achieved on the SDC-OPR system, we next assessed its performance across different FOVs (for high-throughput experiments) and penetration depths. Figure 4a displays a DNA-PAINT image of the microtubule network in fixed HeLa cells, labeled with primary DNA-conjugated antibodies targeting alpha-tubulin. Captured at the SDC-OPR's lowest magnification (1×, 211 × 211 μm² FOV), this image illustrates high-resolution SMLM imaging (average $\sigma_{SMLM}$ of 9.5 nm, Fig. 4a, lower panel) across multiple cells in a single acquisition.

To evaluate the uniformity of localization precision across the FOV, we analyzed 10-μm-thick radial segments at increasing distances from the center in DNA-PAINT images of microtubules acquired with the SDC-OPR at different magnifications (1×, 2.8×, and 4×, Supplementary Fig. 7). At 2.8× and 4× magnifications, localization precision showed minimal radial degradation, increasing by less than 1 nm—from an $\sigma_{SMLM}$ of 1.9 nm at the center to 2.6 nm at the edge of the 76 × 76 μm² FOV. At 1× magnification, this level of degradation for $\sigma_{SMLM}$ remained limited to ~120 × 120 μm² ($\sigma_{SMLM}$ of 6.5 nm at the center to 7.8 nm at the edge). However, across the full 211 × 211 μm²

FOV, localization precision nearly doubled, increasing from 6.5 nm at the center to 12.4 nm at the edge. This effect is attributed to the Gaussian profile of the beam illuminating the excitation spinning disk, which leads to reduced excitation at the edges of larger FOVs. Since localization precision in SMLM is inversely proportional to the square root of detected photons, variations in precision across different magnifications and FOV sizes are expected. Nevertheless, achieving an average $\sigma_{SMLM}$ of 9.5 nm (Fig. 4a, lower panel) over the extended 211 × 211 μm² FOV is remarkable for confocal-based systems, facilitating the exploration of biological heterogeneity at high-resolution. An example of such heterogeneity-related questions addressed using this approach is provided in Supplementary Fig. 8, where we examined the spatiotemporal organization of the T cell receptor internal zeta chain (TCRζ) within Jurkat T cells.

For $\sigma_{SMLM}$ depth-dependent analysis, we next acquired DNA-PAINT images of the microtubule network in fixed HeLa cells of a confocal volume of ~500 nm thickness using the system highest magnification (4×, 53 × 53 μm² FOV). This volume was sequentially imaged in 1-μm steps throughout the cell's 9 μm height (Fig. 4b and Supplementary Fig. 9). Figure 4c displays the dependence of the distribution of $\sigma_{SMLM}$ with penetration depth. Notably, as depth increases, the number of detected photons diminishes due to scattering and optical aberrations, resulting in reduced localization precision for both the localization precisions derived from CRLB along with those calculated using the NeNA metric (see Supplementary Fig. 9). Still, the high level of photon collection possible by the SDC-OPR system, enables $\sigma_{SMLM} \leq 10$ nm for up to 9 μm imaging depth with narrow distributions. Indeed, double-walled filamentous microtubule structures were clearly resolved at various depths: near the coverslip–cell interface, at intermediate axial positions, and at the top of the cell, with peak-to-peak distance between 30 and 40 nm (Fig. 4d), consistent with reported values[24]. These results highlight the SDC-OPR's ability to deliver high-resolution imaging across large FOVs and throughout the entire height of cells.

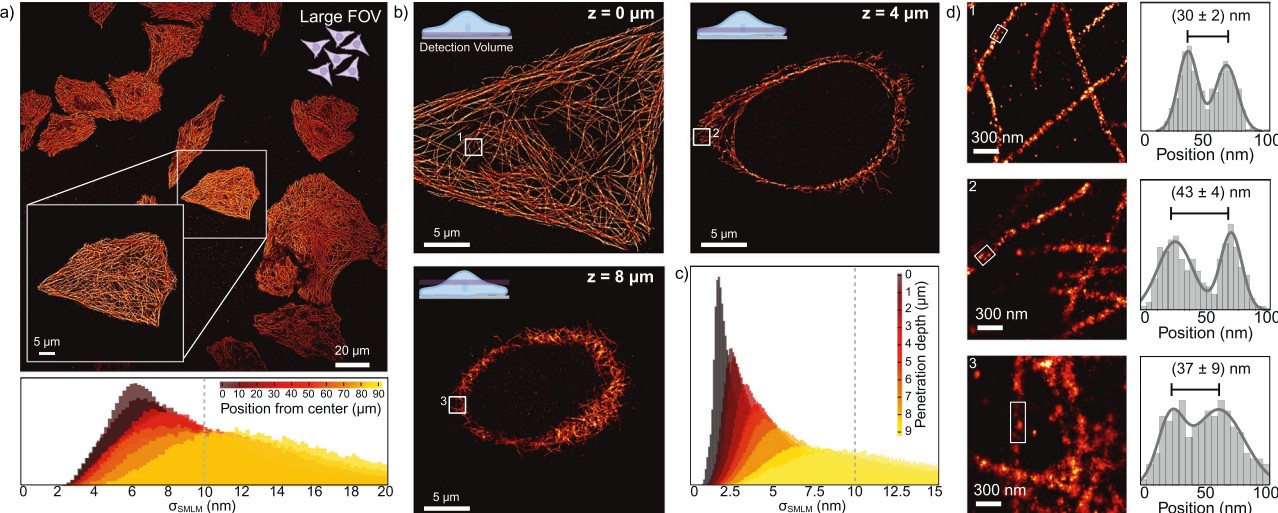

**Fig. 4 | Whole cell DNA-PAINT imaging within large field-of-views on an SDC-OPR microscope. a** Top: SDC-OPR DNA-PAINT image of microtubules in HeLa cells across a large FOV of 211 × 211 µm². Cells were stained with DNA-labeled anti-alpha-tubulin antibodies, as described in the Methods section. Bottom: Distribution of localization precision ($\sigma_{SMLM}$) measured at different distances (in µm) from the center of the FOV, with each color representing a different distance. **b** SDC-OPR DNA-PAINT images of the microtubules at different penetration depths (0, 4, and 8 µm) within HeLa cells. A cartoon in the upper left corner provides a schematic representation of each penetration depth relative to the overall cell volume.

**c** Distribution of $\sigma_{SMLM}$ for each penetration depth for microtubules sample at 4× magnification, with each color representing a different penetration depth. **d** *Left*: Zoomed-in images of highlighted small regions from each penetration depth indicated in (**b**). Right: Position distribution for the highlighted regions in the lefthand panels, quantifying the distances between microtubule walls in nanometers. Microtubule imaging was repeated in three independent experiments. Source data of histograms are provided as a Source Data file. Schemes of (**a**, **b**) were created with BioRender.com.

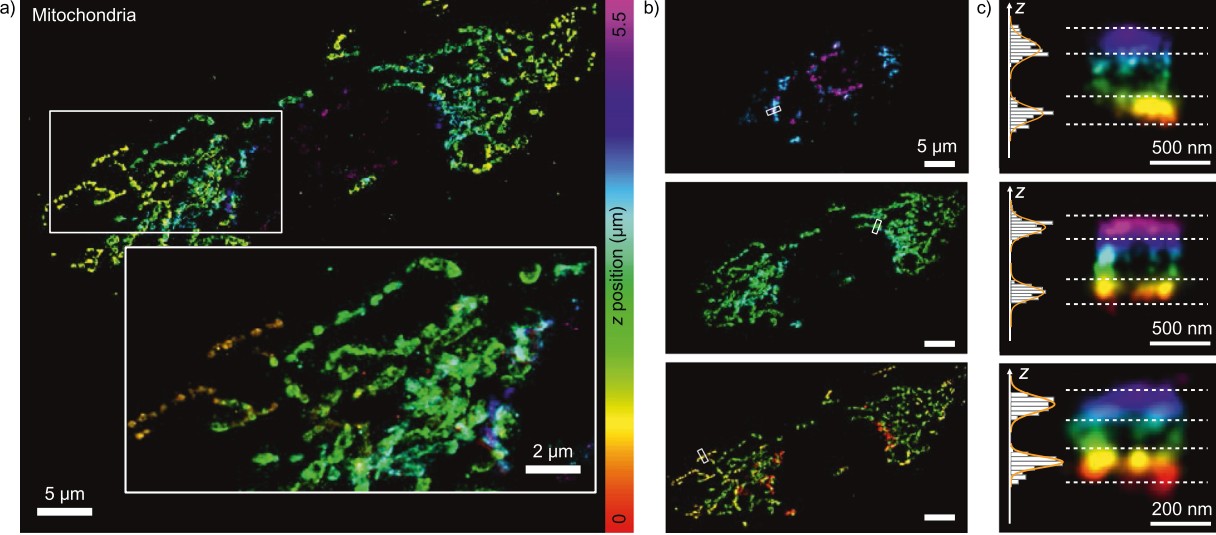

**Fig. 5 | 3D whole cell DNA-PAINT imaging using an SDC-OPR microscope. a** 3D DNA-PAINT image of mitochondrial structures in U2OS cells acquired with an SDC-OPR microscope. Mitochondria were labeled using a DNA-conjugated secondary nanobody targeting a primary antibody against TOM20. Details on sample preparation are provided in the Methods, and imaging conditions are summarized in Supplementary Table 1. Color indicates axial position. A zoomed-in region highlights the mitochondrial network structure. **b** Differential axial sections from (**a**) at distinct axial ranges: 0–1.8 µm (top), 1.8–3.6 µm (middle), and 3.6–5.5 µm (bottom).

The color scale is consistent with (**a**). Scale bars = 5 µm. **c** Cross-sectional views (*yz* or *xz*) from individual mitochondria at different axial ranges denoted by white dashed lines in (**b**). Axial histograms depict mitochondrial membrane distributions, with standard deviations of 30 nm and 42 nm (top), 50 nm and 51 nm (middle), and 51 nm and 65 nm (bottom). Bead calibration using cspline 2D model algorithm is detailed in Supplementary Fig. 10. 3D mitochondria sample imaging was repeated in three independent experiments. Source data of histograms are provided as a Source Data file.

## Whole cell 3D super-resolution imaging via DNA-PAINT on an SDC-OPR system

To evaluate the capabilities of 3D DNA-PAINT imaging on an SDC-OPR system, we visualized the intricate mitochondrial network in U2OS cells by sequentially capturing 500-nm-thick z-sections across the entire 5.5 µm cell height (4×, 53 × 53 µm² FOV). Each section was processed using the single-molecule fitting algorithm for arbitrary PSF

models developed by the Ries lab[25], allowing for precise 3D reconstruction from 2D images without additional optical modifications to the SDC-OPR system. Details on calibration and localization precision as a function of z-position within a single depth of field are provided in Supplementary Fig. 10.

Figure 5a presents the z-color-coded whole cell image of the mitochondrial network, offering a detailed visualization of its 3D

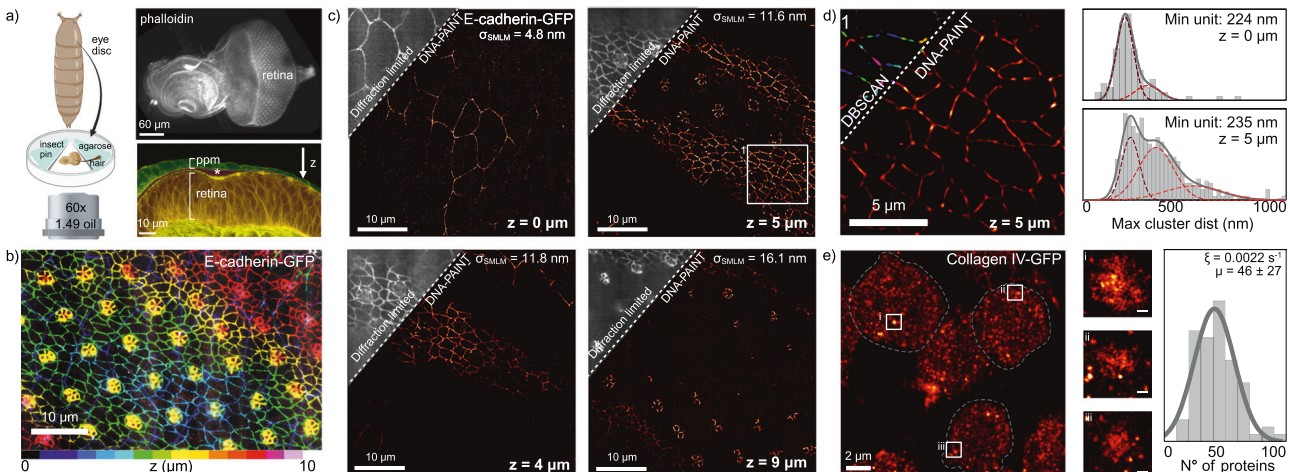

**Fig. 6 | DNA-PAINT imaging of *Drosophila* retinal epithelium on an SDC-OPR microscope up to 9 μm penetration depth. a** Left: Schematic of 3rd instar *Drosophila* eye disc and sample mounting for SDC-OPR imaging, described in Methods. Right: Phalloidin-stained eye disc (top). The peripodial membrane (ppm), luminal space (*), and retinal epithelium (retina) are highlighted with green, magenta, and orange masks, respectively (bottom). **b** SPC-OPR 3D projection of E-cadherin-GFP with 0.5 μm z-step, showing 10 μm depth with color-coded axial position. **c** Representative SDC-OPR DNA-PAINT images of E-cadherin at different penetration depths (0, 4, 5, and 9 μm). E-cadherin-GFP was labeled with DNA-conjugated anti-GFP antibodies (see Methods). Panels include localization precision ($\sigma_{SMLM}$) for each penetration depth. **d** Zoomed-in view of the highlighted region from (**c**), showing density-based clustering analysis (DBSCAN) of the E-cadherin DNA-PAINT image. Histogram displays the maximum distances within clusters, derived from the sample shown in (**c**) for $z = 0$ μm and $z = 5$ μm, and the value of the smallest pick of the histogram (Min unit). **e** Collagen-IV imaging of the basement membrane showing three example cells outlined (full image can be found in Supplementary Fig. 14). $\sigma_{SMLM}$: 4.6 nm. Collagen-IV is labeled with GFP and DNA-labeled anti-GFP nanobody was used for DNA-PAINT imaging. Central panels i, ii and iii provide zoomed views of three examples vesicles. The histogram quantifies the number of Collagen IV molecules present in all analyzed vesicles, with qPAINT quantification indicating an average (μ) of $46 \pm 27$ molecules per vesicle. $\xi$ is the qPAINT index used for calibration. See methods. Scale bar: 100 nm. All *Drosophila* imaging was repeated in three independent experiments. Source data of histograms are provided as a Source Data file. Schematic of panel (a) was created with BioRender.com.

organization. Selected views within a 1.8 μm axial range are shown in Fig. 5b. Cross-sectional analysis of mitochondria at different depths revealed TOM20 localization at the outer mitochondrial membrane, demonstrating the method's ability to resolve the hollow structure of mitochondria (Fig. 5c). To assess z-resolution across different depths, we measured the thickness of these individual mitochondrial membranes. The mean membrane thickness across the entire cell was $48 \pm 12$ nm, with a minimum of ca. 30 nm for mitochondria located near the coverslip (Fig. 5c, side views). This suggests a slight degradation in z-precision with increasing imaging depth, consistent with the observed lateral resolution decline at greater depths. Still, this range of values agrees well with reported values for mitochondrial wall thickness by other super-resolution imaging implementations and is consistent with the expected dimensions of mitochondrial membranes (~7.5 nm, as measured by cryo-electron tomography)[26], the combined sizes of primary and secondary antibodies (~20 nm), and the CRLB axial localization error (~20 nm, Supplementary Fig. 10). Altogether, these results highlight the capability of 3D DNA-PAINT on an SDC-OPR system to achieve high-resolution imaging across cellular depths, demonstrating its potential for detailed structural studies in complex biological systems.

## SDC-OPR combined with DNA-PAINT reveals heterogenous distribution of E-cadherin at the AJ in the *Drosophila* retinal epithelium

To showcase the power and ease of SMLM imaging on an SDC-OPR system for biology research, we investigated the spatial distribution of GFP-tagged E-cadherin[27] at the retinal cell AJs, in the developing *Drosophila* eye imaginal disc. Conducting SMLM studies in intact tissues remains a significant challenge. Localization recovery at greater depths is typically restricted to experiments within isolated cells or first cellular layer in multicellular samples, or to investigations of transformed tissue[28,29].

The eye imaginal disc consists of a thin squamous epithelium, called the peripodial membrane (ppm), that overlies the apical surface

of the columnar retinal epithelium (Fig. 6a). The AJs mediate cell-cell adhesion and are located at the border between the apical and lateral domains of the cells. To image the retinal epithelium, third instar eye discs were mounted in a modified glass-bottomed dish (Supplementary Fig. 11), with the ppm positioned facing the objective lens. This setup required imaging through both the ppm and the luminal space to visualize the underlying retinal epithelium. For DNA-PAINT imaging, the samples were immunostained using DNA-conjugated anti-GFP nanobodies, enabling precise detection of GFP-tagged E-cadherin at super-resolution.

Diffraction limited SDC-OPR imaging using the GFP channel, which highlights GFP-tagged E-cadherin, reveals continuous staining at the level of the AJs in the ppm (Fig. 6b). As previously reported, E-cadherin is enriched between the newly differentiated photoreceptor neurons and is expressed at lower levels between the remaining progenitor cells that surround the photoreceptors[30]. DNA-PAINT imaging on the SDC-OPR allowed to capture high-resolution images of E-cadherin with an average $\sigma_{SMLM}$ of $(12 \pm 1)$ nm over a 9 μm depth (Fig. 6c). This revealed a heterogeneous distribution of E-cadherin, with distinct high-intensity foci observed between cells in both the squamous ppm epithelium ($z = 0$ μm) and the underlying columnar retinal epithelium ($z > 0$ μm) (Fig. 6c). This represents the first demonstration of size heterogeneity at nanometer resolution in *Drosophila* retinal development, echoing similar findings in the developing embryonic epidermis[31]. To further investigate the size distribution of E-cadherin foci in both epithelial layers, we applied density-based clustering analysis (DBSCAN) to DNA-PAINT data at $z = 0$ μm (squamous peripodial cells) and $z = 5$ μm (progenitor columnar cells of the eye disc). This analysis identified a minimum E-cadherin foci size of $(230 \pm 40)$ nm across both cell types. However, columnar cells exhibited larger foci domains, reaching $(352 \pm 90)$ nm and $(470 \pm 160)$ nm, corresponding to 1.5× and 2× the size of the smallest foci, respectively (Fig. 6d) (see Supplementary Fig. 12 for line profile comparisons of GFP vs DNA-PAINT images at 5 μm depth). These quantitative measurements provide valuable insights into the spatial organization of

E-cadherin at the AJ of a developing epithelium by highlighting that these cell contacts are heterogeneous and present nanoclusters of finite size. This finding is consistent with what has been reported before in the fly embryo using PALM imaging under HILO illumination, with E-cadherin domains ranging from 200 nm to 600 nm in length[32]. Notably, our study extends these previous findings by demonstrating that this nanometer-scale organization is conserved across distinct epithelial tissues, suggesting that the minimal E-cadherin foci size may represent a fundamental property of E-cadherin-mediated adhesion. Importantly, our imaging approach extends this previous finding by enabling the examination of E-cadherin organization at greater penetration depths in more complex developing tissues.

To further assess the capability of DNA-PAINT imaging at greater penetration depths, we extended our imaging to the pupal fly retina, where photoreceptors elongate along the lens-to-brain axis and their AJs align accordingly, providing an opportunity to study imaging performance at increased depth in a tissue context. Unlike the third instar eye disc, which is a thinner, curved epithelial sheet, the pupal retina features a more stratified organization with deeper cellular layers, presenting a more demanding test for high-resolution imaging. Remarkably, we were able to super-resolve photoreceptor AJs at depths of up to 15 μm with a $\sigma_{SMLM}$ of $(26 \pm 1)$ nm and a NeNA of 21.6 nm (see Supplementary Fig. 13). These results underscore the ability of the SDC-OPR system to achieve high-resolution imaging through multiple cell layers and across large fields-of-view, offering significant advantages over conventional TIRF or HILO excitation.

### Quantitative super-resolution imaging of Collagen-IV vesicles in the developing retina

To further test our imaging method, we applied it to the developing retina to image Collagen-IV, a key component of the basement membrane (BM); the extracellular matrix that lines the basal (bottom) surface of epithelia. At larva stages, Collagen-IV is produced by the fat body and diffuses to the basal surface of epithelia. In addition, circulating hemocytes also contribute to depositing Collagen-IV at this tissue surface[33].

Using a strain where Collagen-IV is endogenously labeled with GFP (GFP::Collagen-IV), we were able to visualize intracellular vesicles containing Collagen-IV in hemocytes localized at the basal surface of the developing retina (Fig. 6e and Supplementary Fig. 14). To quantify the number of Collagen-IV molecules within these vesicles, we employed quantitative DNA-PAINT (qPAINT) analysis[34–38]. qPAINT relies on the analysis of binding kinetics between imager and docking strands, specifically by measuring the average dark time (i.e., the waiting time between binding events) of a cluster of single-molecule localizations, as depicted in Supplementary Fig. 15a. The inverse of the measured dark time, known as the influx rate ($\xi$) or qPAINT index, is directly proportional to the number of proteins within that cluster, enabling robust molecular quantification.

Using qPAINT analysis, we determined that each vesicle contained an average of $46 \pm 27$ Collagen-IV molecules (Fig. 6e). As previously reported, Collagen-IV vesicles measure approximately 300 nm in all dimensions, and imaging a single focal plane was sufficient to capture their full axial extent[33,39–42]. Importantly, qPAINT-based molecular quantification relies on internal calibration within the same sample, leveraging the detection of single Collagen-IV molecules within the FOV. This approach has proven to be highly reproducible across independent experiments, as demonstrated by two independent replicates presented in Supplementary Fig. 15b, further underscoring the robustness of this method for precise molecular quantification in a tissue context (Supplementary Fig. 15).

These results emphasize the transformative potential of DNA-PAINT on SDC-OPR to explore subcellular structures in a tissue context, offering remarkable resolution and quantitative precision.

## Discussion

Advancements in fluorescence microscopy have become pivotal in life sciences, offering the ability to unravel intricate biological processes with exceptional detail. However, traditional microscopy techniques often fall short in resolving subcellular components, particularly when investigating protein organization at the nanoscale. While super-resolution fluorescence microscopy has significantly enhanced spatial resolution by surpassing the diffraction limit, challenges persist that hinder its widespread application in biological research. Key among these challenges are the limitations in FOV, resolution, and penetration depth, which are often difficult to optimize simultaneously. Among the most impressive SMLM implementations for high-resolution in-depth imaging, 4PI microscopy reaches ~10–20-nm isotropic resolution[43] at depths of up to 9 μm but is constrained by a limited FOV of $17 \times 17$ μm². Similarly, modulated excitation microscopy can reach $\sigma_{SMLM}$ values as low as 3 nm over 7 μm in depth, yet only within a small FOV of just $15 \times 15$ μm² [44]. Furthermore, these techniques demand sophisticated instrumentation and expert knowledge, making them less accessible for most biological studies. This gap has driven demand for versatile and accessible super-resolution microscopy methods that can bridge these limitations, enabling researchers to explore complex tissue environments with unparalleled detail.

To address these challenges, we demonstrated the potential of combining SMLM with spinning disk confocal microscopy enhanced by optical photon reassignment (SDC-OPR). By integrating microlens arrays into a traditional SDC setup, the commercially available SDC-OPR configuration increases photon collection while reducing the effective pinhole size, significantly improving spatial resolution. Our approach achieved localization precision of sub-2 nm in-plane and sub-10 nm up to 9 μm depth, all within a significantly larger and adjustable FOV of $53 \times 53$ μm² or $76 \times 76$ μm², highlighting its capacity to deliver high-resolution imaging through whole cells or tissue samples without sacrificing accuracy and without the need of complex instrumentation. Notably, our findings emphasize the necessity of using the OPR reassignment unit to achieve superior resolution in DNA-PAINT imaging, compared to simply increasing laser power in a standard SDC setup. While increased laser power can enhance photon emission, it also introduces risks of phototoxicity, photobleaching, and photo-induced depletion of DNA-docking sites[45], all of which impose practical limits on resolution improvement. The OPR unit overcomes these limitations by optimizing photon reassignment, enabling higher photon collection efficiency without additional stress on the sample[46]. This capability is demonstrated in Fig. 1d, where the SDC-OPR system resolves 10 nm docking sites that remain unresolvable with the SDC under identical power conditions (Fig. 1c). This approach provides a robust and efficient means to achieve sub-10 nanometer-scale resolution required for protein-scale imaging in biological systems.

Beyond high-resolution single-color imaging, the SDC-OPR system enables multicolor SMLM experiments, which are essential for studying cellular structures and interactions. In this study, we demonstrate its multicolor imaging capabilities by sequentially imaging Nup96-mGFP, mitochondria and α-tubulin in U2OS cells using orthogonal DNA imager strands, leveraging DNA-PAINT's exchange mechanism for multiplexed imaging. Additionally, we applied RESI imaging[6], which utilizes Exchange-PAINT, to visualize Nup96-mGFP in U2OS cells with 3 Å localization precision. These results highlight the adaptability of the SDC-OPR system for high-precision multiplexed imaging, further expanding its utility in nanoscale biological investigations. Notably, beyond Exchange-PAINT, multicolor imaging can also be achieved through simultaneous two-color acquisition using the dual-camera configuration commonly available in commercial SDC-OPR systems. This approach can enable real-time acquisition of two spectrally distinct channels, enhancing the potential for multiplexed imaging applications. For such implementations, the recently identified best-performing dyes for 488 nm, 560 nm, and 640 nm

excitation[47] could facilitate a combination of sequential and spectral separation strategies within the SDC-OPR configuration, enabling highly multiplexed imaging.

A particularly impressive feature of DNA-PAINT imaging on the SDC-OPR system is its scalability. When the FOV was expanded to a much larger area ($211 \times 211\,\mu m^2$), DNA-PAINT imaging maintained a high localization precision of sub-10 nm in-plane. This scalability is crucial for biological applications, where large tissue areas or numerous cells need to be imaged with nanoscale precision to capture essential subcellular details and produce statistically robust datasets − a significant limitation in many other super-resolution approaches, which sacrifice FOV to achieve high-resolution imaging.

3D DNA-PAINT imaging on the SDC-OPR platform, combined with the arbitrary PSF models developed by the Ries lab[25], enabled high-resolution visualization of the mitochondrial network in cells without requiring additional optical elements to modify the PSF. By capturing 500-nm-thick z-sections across a $5.5\,\mu m$ cell height, the system successfully reconstructed the mitochondrial network, resolving the hollow structure of mitochondria and precisely mapping TOM20 at the outer mitochondrial membrane in 3D. Notably, its axial resolution is comparable to that achieved with DNA-PAINT imaging on an SDC equipped with astigmatic lenses[14]. While incorporating astigmatic lenses into commercially available SDC-OPR systems could only further enhance 3D volumetric imaging by improving depth precision, the current system already demonstrates good performance in resolving fine mitochondrial structures at comparable axial resolutions to other SMLM in depth implementations[48].

Besides benchmarking, our imaging approach unveiled previously unseen E-cadherin nanodomains present at the AJ of the retinal and peripodial cells. Specifically, DNA-PAINT imaging on an SDC-OPR allowed imaging of the AJ and collagen deposition in a developing epithelium, achieving $\sigma_{SMLM}$ ~12 nm across multiple cell layers over a $9\,\mu m$ depth.

Notably, we observed previously undetected heterogeneity in the distribution of E-cadherin at retinal AJs. Previous work using PALM in the fly epidermis revealed in this developing tissue that the AJ consists of two populations of E-cadherin, with domains where this adhesion receptor is concentrated and domain where it is found at lower levels. Our imaging approach, bypassing TIRF penetration depth, now shows that this type of bipartite distribution extends to two other developing epithelia, suggesting it is a feature that is common to all developing tissues[32].

To further showcase our imaging approach, we imaged Collagen-IV deposition during retinal development. Collagen-IV is deposited by the hemocytes which patrol the basal surface of the retinal epithelium. Our methods allowed us to count the number of Collagen-IV molecules within hemocyte vesicles underscoring its potential for precise molecular quantification, even in tissue samples. Together, these findings validate that DNA-PAINT on an SDC-OPR is a powerful and accessible tool for studying subcellular structures and molecular distributions with remarkable accuracy.

Building on this foundation, we anticipate that the ability to achieve such high-resolution imaging at depth will open new avenues for investigating a wide range of biological questions, including the dynamics of cellular adhesion, tissue morphogenesis, and extracellular matrix organization in various developmental contexts. Recent advancements in DNA-PAINT imaging have introduced dye-quencher or dye-dye-based self-quenching imager probes, which significantly reduce background fluorescence, enhance brightness, and improve spatial resolution while achieving faster imaging speeds[49–51]. Incorporating these fluorogenic probes within the SDC-OPR system could further amplify imaging speed and resolution. By marrying these emerging innovations with the capabilities of the SDC-OPR platform, future research could address even more complex biological systems, making this approach a versatile tool for studying cellular and molecular architecture across diverse tissue types.

In conclusion, the combination of SDC-OPR with DNA-PAINT not only enhances spatial resolution and quantitative imaging but also expands the scope of super-resolution microscopy to a wider range of biological contexts, from cell cultures to complex tissue environments. The ability to integrate DNA-PAINT imaging with existing commercially available SDC-OPR makes it an appealing option for broader adoption in the biological research community. This advanced approach opens opportunities for imaging in a wider range of biological contexts, from single-cell analysis to tissue-scale studies, with the potential to drive significant advancements in our understanding of tissue development, molecular signaling, and structural organization at the nanoscale in health and disease. While our study demonstrates the power of photon reassignment microscopy for tissue super-resolution imaging, it remains an open question whether other high-resolution techniques, such as instant structure illumination microscopy (SIM) or multi-view SIM, could achieve similar levels of resolution while enabling even deeper imaging. Given their capacity for structured illumination and multi-angle acquisition, these approaches could, in principle, complement or extend the depth penetration achieved with SDC-OPR while preserving high spatial precision. Future studies comparing these methodologies will be essential to continue pushing the limits of super-resolution microscopy in thick and complex biological samples.

## Methods
### DNA origami synthesis
The rectangular DNA origami structure with a pillar in the center (Fig. 1b) was designed using CaDNAno[52] and it is accessible at https://nanobase.org/structure/146 [53]. It is modified with six biotin staples going out of the structure for binding to the surface. For the DNA origami with edge pairs separated by 10 nm, 3 pairs of DNA PAINT docking strands (TCCTCCTCCTCCTCCTCCT and ACACACACACACACACA) and two fixed dyes (ATTO 532 and ATTO 647 N) at two ends of the structure were included. For the DNA origami with edge pairs separated by 6 nm, 2 pairs of DNA PAINT docking strands (CTCTCTCTCTCTCTCTCTC and ACACACACACACACACACA) were used. It is based on a 7249-nucleotide long scaffold extracted from the M13mp18 bacteriophage (Tilibit Nanosystems GmbH) and folded into the desired shape using 243 staples folded in 1× TAE (Alfa Aesar, #J63931) and 12 mM $MgCl_2$ (Alfa Aesar, #J61014) buffer. It was mixed in a 10-fold excess of staples over scaffold, and 100-fold for especially modified staples.

Unmodified staple strands were purchased from IDT; biotin-functionalized staples, dye-labeled staple strands with ATTO 532 and ATTO 647 N as well as DNA-PAINT docking strands were purchased from Biomers GmbH. The fluorophores used here are linked through a C6-linker to the single-stranded DNA to the 3′-end. The scaffold and the staple mix were self-assembled into the designed structure using a temperature ramp. The combination was initially heated to 95 C°, where it remained for 5 min, before being cooled to 20 C° during a 19 h linear ramp. A 1% agarose gel electrophoresis was used as a purification procedure to remove excess staple strands. In a 1× TAE 12 mM $MgCl_2$ buffer, the gel was run at 70 V for 3 h in an ice-water bath. After electrophoresis, the pure DNA origami structures were extracted by cutting out the bands in the gel containing the DNA origami structure and squeezing them out using a parafilm-wrapped glass slide. The final concentration of the DNA origami structures was determined using a Nanodrop 2000 spectrophotometer (Thermo Fisher Scientific). The list of modified staples used can be found in Supplementary Table 2.

### Buffers
Buffer A + : 10 mM Tris (VWR) pH 8.0, 100 mM NaCl (Sigma-Aldrich) and 0.05% Tween-20 (Sigma-Aldrich).

Buffer B + : 10 mM MgCl$_2$ (LifeTech), 5 mM Tris-HCl pH 8.0, 1 mM EDTA (LifeTech) and 0.05% Tween-20 (Sigma-Aldrich) pH 8.0, optionally supplemented with 1× trolox, 1× PCA and 1× PCD.

Buffer C + : 1× PBS, 1 mM EDTA (Invitrogen), 500 mM NaCl (Sigma-Aldrich) pH 7.4, 0.02% Tween (Sigma-Aldrich), optionally supplemented with 1× trolox, 1× PCA and 1× PCD.

PCA, PCD and Trolox: To summarize, this system consisted of 1× protocatechuic acid (PCA, stock 40× solution), 1× protocatechuate 3,4-dioxygenase (PCD, stock 100× solution) and 1× ( ± )-6-Hydroxy-2,5,7,8-tetramethylchromane-2-carboxylic acid (Trolox, stock 100× solution) in 1× PBS + 500 mM NaCl buffer and incubated in the dark for 1 h before imaging. 40× PCA stock was made from 154 mg of PCA (Sigma-Aldrich) in 10 ml of distilled water adjusted to pH 9.0 with NaOH (Avantor). 100× PCD solution was made by adding 2.2 mg of PCD (Sigma-Aldrich) to 3.4 ml of 50% glycerol (Sigma-Aldrich) with 50 mM KCl (Sigma-Aldrich), 1 mM EDTA (Invitrogen) and 100 mM Tris buffer (VWR). 100× Trolox solution was made by dissolving 100 mg of Trolox (Sigma-Aldrich) in 0.43 ml methanol (Sigma-Aldrich), 0.345 ml 1 M NaOH and 3.2 ml of distilled water.

Cytoskeleton Buffer (CB): 10 mM MES (Sigma-Aldrich), pH 6.1, 150 mM NaCl (Sigma-Aldrich), 5 mM EGTA (Sigma-Aldrich), 5 mM D-glucose (LifeTech), 5 mM MgCl$_2$ (LifeTech); described in ref. 54.

## DNA origami sample preparation
For sample immobilization, commercial chambers (IB-80607 | µ-Slide VI 0.5 Glass Bottom, Sterile) were used. The surface was passivated with BSA biotin (1 mg/mL$^{-1}$, A4503-10G, SigmaAldrich) for 20 min at room temperature on top of a rotating platform. After 3 washes with buffer A +, slides were incubated with neutravidin (1 mg/mL$^{-1}$, no. 31,000 Thermo Fisher Scientific) diluted in buffer A+ and incubated for 20 min at room temperature on a rotation platform. After an additional wash with buffer B +, 100 pM of DNA origami structure was added to the chamber and incubated for 15 min for immobilization via biotin binding to the functionalized surface. The sample was washed again with buffer B+ and 100 µl of gold nanoparticles (90 nm, no. G-90-100, Cytodiagnostics) was flushed through and incubated for 5 min before washing with buffer B +. Finally, 180 µl of imager solution in buffer B+ supplemented with 1× trolox, 1× PCA and 1× PCD was flushed into the chamber for imaging.

## Antibody–DNA conjugation
An antibody against alpha-tubulin (MA1-80017 (YL1/2), Thermo Fisher Scientific) was conjugated to DNA-PAINT docking strand 'R2:5'-Thiol-AAACCACCACCACCACCACCA-3' (Custom, Eurofins) via maleimidePEG2-succinimidyl ester coupling reaction. In brief, 1 mM thiolate DNA docking strand was reduced with freshly prepared 250 mM DTT (Thermo Fisher Scientific) solution for 2 h at room temperature. The alpha-tubulin antibody was then concentrated using 100 kDa Amicon spin filter (Merck/EMD Millipore) before mixing with 20× molar excess of maleimide-PEG2-succinimidyl ester cross-linker (Sigma-Aldrich) for 90 min at 4 °C in the dark. To remove excess DTT and cross-linker, the reactions were purified by spin filtration using a Microspin Illustra G-25 column (GE Healthcare) and a Zeba spin desalting column (7 K MWCO, Thermo Fisher Scientific), respectively. The reduced DNA docking strand was added to the purified alpha-tubulin-crosslinker solution at 10× molar excess and incubated on a shaker overnight at 4 °C in the dark. Finally, excess DNA was removed from the alpha-tubulin product via 100 kDa Amicon spin (Merck/EMD Millipore) filtration and stored at 4 °C. Antibody-DNA concentration was measured using a NanoDrop One spectrophotometer (Thermo Fisher Scientific), with the final ratio of DNA: antibody measured to be 1.3. This method was repeated for coupling of an antibody against TCRζ (6B10.2, 644102, BioLegend) to the DNA-PAINT docking strand 'R3:5'Thiol- ACACACACACACACACACA-3' (Custom, Eurofins). The final ratio of DNA: antibody was measured to be 1:1.2.

## Cell culture
HeLa Kyoto mEGFP-Nup107 were cultured in Dulbecco's Modified Eagle Medium (DMEM) supplemented with 10% Fetal Bovine Serum (FBS) and 1% Penicillin with Streptomycin (P/S). Cells were maintained at $2 \times 10^5$ cells/ml and passaged using trypsin-EDTA.

U2OS CRISPR mEGFP-Nup96 cells were cultured in McCoy's 5 A media (ThermoFisher, 16600082) supplemented with 10% FBS and 1% P/S. Cells were maintained at $1 \times 10^5$ cells/ml and passaged using Accutase (Promo Cell, C-41310).

Jurkat E6.1 T cells were cultured in Roswell Park Memorial Institute (RPMI) supplemented with 10% FBS and 1% P/S. Cells were maintained at $1 \times 10^5$ cells/ml.

## Nup96 EGFP, mitochondria and alpha-tubulin labeling in U2OS CRISPR mEGFP-Nup96 cells and HeLa Cells
24 hours prior to imaging $3 \times 10^4$ ($1 \times 10^5$ cells/ml) U2OS CRISPR mEGFP-Nup96 cells were seeded in 8 well Ibidi microslides (IB-80807) in McCoy's 5 A media as described above.

For Nup96 imaging, media was then exchanged with phosphate buffered saline (PBS) and cells were fixed using 4% paraformaldehyde (PFA) for 30 min. Permeabilization was then achieved via addition of 0.1% Triton X-100 for 5 min. Samples were washed 3 × 3 min in 60 mM Glycine in PBS to quench autofluorescence. Blocking was done through the addition of 5% Bovine Serum Albumin (BSA) for 60 min in the dark. 20 nM DNA coupled anti-GFP nanobody (Massive-sdAB-FAST 2-Plex, Massive Photonics GmbH) in 5% BSA was added for one hour at room temperature. Samples were washed 3 × 3 min with PBS before the addition of 90 nm gold nanoparticles (no. G-90-100, Cytodiagnostics) for 5 min. For imaging, 1 nM DNA imager solution (F3: Cy3B from Massive-sdAB-FAST 2-Plex, Massive Photonics GmbH) with 3' attached Cy3b fluorophore in C+ buffer supplemented with 1× Trolox, 1× PCA and 1× PCD was used.

For mitochondria and microtubule imaging the protocol was adapted from ref. 25. After the overnight incubation, most of the cell culture medium was aspirated using a glass pipette placed carefully into a corner of each chamber. Simultaneously, 200 µl of 0.3% (v/v) glutaraldehyde in Cytoskeleton Buffer (CB) + 0.25% (v/v) Triton X-100 is added down the side of the chamber using a second pipette. This prefixation solution is kept for 2 min. Then, cells are fixed with 2% (v/v) glutaraldehyde in CB for 10 min. Auto-fluorescence was quenched with 3 × 3 min of Glycine 60 mM in PBS. Cells were then blocked prior to staining using 0.5 mg ml$^{-1}$ of sheared salmon sperm DNA (Thermo-Fisher, 15632011) in 3% BSA in PBS for one hour. Then, 5 µg/ml of the coupled anti-alpha-tubulin antibody (described in Antibody–DNA conjugation section) and/or 1:200 dilution of TOM20 antibody (Ab186735, abcam) in 2% BSA was added for 2 hours at room temperature. Samples were washed 3 × 3 min with PBS. For mitochondria DNA-PAINT imaging, 25 nM of DNA F2 conjugated anti-rabbit Fab (Massive Photonics) was incubated for 1 h at RT, before washing 3 × 3 min with PBS.

For microtubule imaging, 1 nM DNA imager solution (R2: 5'-TGGTGGT-3') 3' attached Cy3b fluorophore in C+ buffer was used.

For mitochondria imaging, 1 nM DNA imager solution (F2: Cy3B from Massive-sdAB-FAST 2-Plex, Massive Photonics GmbH) with 3' attached Cy3b fluorophore in C+ buffer supplemented with 1× Trolox, 1× PCA and 1× PCD was used.

For Exchange-PAINT imaging of Nup96, microtubules, and mitochondria, the immunostaining protocol described above for microtubule and mitochondria imaging was followed. Before adding the corresponding DNA imager solution, the imager solutions were exchanged by thoroughly washing with C+ buffer.

## TCRζ labeling of activated Jurkat cells
24 h prior to imaging Jurkat T cells were cultured in fresh supplemented RPMI media at $5 \times 10^5$ cells/ml. Glass bottom 6 channel Ibidi µ-

slides (IB-80607) were incubated with 50 µl of activating antibody solution containing 2 µg/ml anti-CD3 (OKT3, 16-0037-85, Invitrogen) and 5 µg/ml anti-CD28 (CD28.2, 16-0289-85, Invitrogen) in PBS overnight at 4 degrees. Fixation, permeabilization, quenching, and blocking steps were then performed as described above. 100 µl of 10 µg/ml anti-TCRζ (6B10.2, 644102, BioLegend) coupled to a DNA docking strand (5′-ACACACACACACACACACA-3′) in 5% BSA in PBS was added and slides were incubated for 1 h at 37 degrees. Slides were washed 3 × 3 min with PBS and 90 nm gold nanoparticles were added for 15 min before washing was repeated. DNA imager solution containing 2 nM DNA imager (5′-TGTGTGT-Cyb3-3′) in C+ buffer was added before imaging.

### Imaging Drosophila third instar eye discs and pupal retina

*Drosophila melanogaster* was used as a model organism in this study. Modified strains BDSC:60584 and BDSC:98343 were obtained from the Bloomington Drosophila Stock Center. Experiments were conducted using third instar larvae and pupae. Tissues (third instar eye-antennal discs and pupal retinae) were dissected from a minimum of five animals per biological replicate. Both male and female animals were included in experiments; however, data were not disaggregated by sex, as no sex-specific effects were anticipated. Ethical approval was not required for work involving *D. melanogaster*. Third instar eye discs and pupal retina were dissected in PBS from transgenic flies where the endogenous E-Cadherin (BDSC 60584)[27] or Collagen-IV (BDSC 98343)[55] loci have been tagged with GFP. Detailed descriptions for the dissection of the third instar eye disk[56] and pupal retina have been previously described[57]. Pupal retinas were staged by selecting pre-pupae that were incubated at 25 °C for 42 h. Dissected eye discs or pupal retinas were fixed in PBS + 4% formaldehyde (Sigma F8775) for 20 min at room temperature, then washed 3× with PBS + 0.3% Triton (Sigma T8787). Samples were blocked with 5% goat serum (MP Biochemicals 642921) in PBS + 0.3% Triton for 20 min at room temperature, then incubated with 20 nM GFP antibody (Anti-GFP Custom docking site (sdAB-5′-TCCTCCTCCTCCT-3′, Massive Photonics GmbH) in PBS + 0.3% Triton overnight at 4 degrees.

Alexa Fluor Plus 405 phalloidin (Thermofisher A30104) was added at 165 nM for 2–4 h at room temperature. Samples were washed 3×, left overnight at 4 °C, then transferred into custom-adapted 35 mm glass-bottomed dishes (Supplementary Fig. 11) and immobilized (see below). Immediately before imaging, the wash was replaced with 10 nM (E-Cadherin) or 5 nM (Collagen-IV) DNA-imager solution (R1: 5′-AGGAGGA-3′) 3′ attached Cy3b fluorophore in C+ buffer supplemented with 1× Trolox, 1× PCA and 1× PCD. Discs and retina were imaged in modified 35 mm glass bottomed dishes. Three insect pins (Fine Science Tools 26002-20) were arranged as a triangle and glued in place over two human hairs using superglue (cyanoacrylate) (Supplementary Fig. 7). A central, triangular chamber with two sections was formed, each having an end of one hair free. 2% agarose was added to the outer sections of the cover glass to minimize the chamber volume. After five minutes, imaging solution was added to the chamber. Two discs/retina were pipetted, each into a section of the triangular chamber, placing each sample under a hair to immobilize them (Fig. 4a).

### Microscopy setups

**TIR setup.** TIR microscopy was carried out on a custom built total internal reflection fluorescence microscope based on a Nikon Eclipse Ti-2 microscope equipped with a 100× oil immersion TIRF objective (Apo TIRF, NA 1.49) and a Perfect Focus System. Samples were imaged under flat-top TIR illumination with a 560 nm laser (MPB Communications, 1 W) magnified with both a custom-built telescope and a variable beam expander, before passing through a beam shaper device (piShaper 6_6_VIS, AdlOptica, Berlin, Germany) to transform the Gaussian profile of the beam into a collimated flat-top profile. Laser

polarization was adjusted to circular using a polarizer followed by a quarter-waveplate. The beam was focused into the back focal plane of the microscope objective using a suitable lens, passed through an excitation filter (FF01-390/482/563/640-25, Semrock) and coupled into the objective using a beam splitter (Di03- R405/488/561/635, Semrock). Fluorescent light was spectrally filtered with an emission filter (FF01-446/523/600/677-25, Semrock) and imaged on a sCMOS camera (Hamamatsu, ORCA-Fusion BT) without further magnification, resulting in a final pixel size of 130 nm in the focal plane, after 2 × 2 binning.

**SCD-OPR setup.** This is a commercial setup CSU-W1 SoRA from Nikon. Multifocal laser excitation is generated by micro-array lenses on the upper disk of the scanning unit. This excitation passes through a corresponding array of pinholes on the lower disk of the scanning unit. The multifocal excitation then scans the specimen through a tube lens and an objective lens. The lower disk also contains micro-array lenses on the underside which focuses the fluorescence emission from the sample by twofold. This displaces emitted photons to their most probable original location, mimicking the effect of an infinitely small pinhole size without compromising brightness. Pinholes further reject out-of-focus emissions to achieve the enhancement in resolution whilst maintaining confocal sectioning. This emission is then reflected by a dichroic mirror and imaged by an ORCA-Fusion BT Digital CMOS camera through magnifying relay optics and emission filters. The SoRA element is added onto a Nikon Ti2 inverted microscope base which has a dual camera function. Magnification on the SDC-OPR can be 1×, 2.8× and 4×, resulting in a pixel size of 108 nm (with no binning, 1× magnification), 78 nm (with 2× binning, 2.8× magnification) and 108 nm (with a 4× binning, 4× magnification).

**SDC.** The spinning disk confocal microscope used for imaging the origamis of top panel of Fig. 1c and Supplementary Fig. 1a, was the same system used for the SDC-OPR measurements (CSU-W1 SoRA from Nikon), but the spinning disk was changed for a normal one instead of using the SoRA disk.

### Imaging parameters

For DNA origami imaging, images were taken on the SCD-OPR (CSU-W1 SoRA Nikon system) using 15,000 frames and an integration time of 300 ms. For top panel of Fig. 1d and the first panel of Figs. 1e, 4 × magnification and 4× binning size were used, giving a final pixel size of 108 nm. The 561 nm laser at 100 % power (1.75 mW measured after the objective) was used in conjunction with the 605/52 bandpass filter. For the second panel of Fig. 1e, 2.8 × magnification and 2× binning size were used, giving a final pixel size of 78 nm. The 561 nm laser at 100% power (3 mW measured after the objective). For the lower panel of Fig. 1e, 1 × magnification and no binning were used, giving a final pixel size of 108 nm. The 561 nm laser at 100% power (5 mW measured after the objective). For the images taken in the TIR setup, DNA origamis were imaged using 20,000 frames and integration time of 100 ms with a power density of 600 W/cm² (Supplementary Fig. 1e).

For Nup96 imaging, images were taken on the SCD-OPR using 17,000 frames and an integration time of 300 ms (Fig. 2a, c and Supplementary Fig. 4a). For Nup96 RESI imaging, images were taken on the SDC-OPR using 17,000 frames for each Exchange-PAINT round and an integration time of 300 ms (Fig. 3b and Supplementary Fig. 6). For Nup96, mitochondria and microtubule Exchange-PAINT imaging, images were taken on the SCD-OPR using 17,000 frames for each round and an integration time of 300 ms (Fig. 3a). For 3D whole cell mitochondria imaging, the entire 5.5 µm height of the cell was imaged sequentially in 0.5-µm steps, acquiring 20,000 frames at 300 ms integration time for each step (Fig. 5). 4× magnification and 4 binning size were used for all these images, giving a final pixel size of 108 nm. The 561 nm laser at 100% power (1.75 mW measured after the

objective) was used in conjunction with the 605/52 bandpass filter. For the images taken in the TIR setup, Nup96 were imaged using 20,000 frames and an integration time of 100 ms at a power density of 160 W/cm$^2$ (Supplementary Fig. 4b).

For microtubule imaging, images were taken on the SCD-OPR (CSU-W1 SoRA Nikon system) microscope using 17,000 frames and an integration time of 300 ms. For the images at different penetration depths (Fig. 4b, Supplementary Fig. 9 and Supplementary Fig. 7c), 4× magnification and 4× binning size were used, giving a final pixel size of 108 nm. The 561 nm laser at 100% power (1.75 mW measured after the objective) was used in conjunction with the 605/52 bandpass filter. For the biggest FOV of Fig. 4a and Supplementary Fig. 7a, 1 × magnification and no binning were used, giving a final pixel size of 108 nm. The 561 nm laser at 100% power (5 mW after the objective) was used in conjunction with the 605/52 bandpass filter. The entire 9 μm height of the cell was imaged sequentially in 1-μm steps. For Supplementary Fig. 7b, 2.8 × magnification and 2× binning size were used, giving a final pixel size of 78 nm. The 561 nm laser at 100% power (3 mW measured after the objective) was used in conjunction with the 605/52 bandpass filter.

For *Drosophila* imaging, images were taken on the SCD-OPR (CSU-W1 SoRA Nikon system) using 18,000 frames and an integration time of 300 ms 4× magnification and 4 binning size were used, giving a final pixel size of 108 nm (Fig. 6 and Supplementary Figs. 12, 13 and 14). The 561 nm laser at 100% power (1.75 mW measured after the objective) was used in conjunction with the 605/52 bandpass filter.

For TCRζ imaging in Jurkat cells (Supplementary Fig. 8), images were taken on the SCD-OPR (CSU-W1 SoRA Nikon system) microscope using 10,000 frames and an integration time of 200 ms. 2.8× magnification and 2× binning size were used giving a final pixel size of 78 nm. The 561 nm laser at 100% power (1.75 mW measured after the objective) was used in conjunction with the 605/52 bandpass filter.

For all imaging conditions on the SDC and SDC-OPR a Nikon 60× Apo NA 1.49 oil immersion lens magnification was used. All imaging parameters can be found in Supplementary Table 1.

## Deconvolution
Deconvolution of the SDC-OPR images was performed using NIS-Elements AR. Each image was processed using the blind deconvolution method with 12 iterations.

## DNA-PAINT analysis
The raw fluorescence videos were processed for super-resolution reconstruction using the Picasso software package[4] (latest version accessible at https://github.com/jungmannlab/picasso). Drift correction was applied using redundant cross-correlation, utilizing gold particles as fiducials for cellular experiments. Typically, around 50 nanoparticles were detected within the field-of-view (FOV), providing reliable drift correction, particularly for lateral shifts at the focal plane ($z = 0$). At higher z-planes, however, gold fiducials were no longer detected. In such cases, drift correction was performed using the adaptive intersection maximization-based method (AIM), a marker-free algorithm developed by Hongqiang Ma et al.[58], which is implemented in the Picasso platform.

Image rendering was carried out using the Render module of Picasso. Each localization was represented as a Gaussian spot, with the spread corresponding to the individual localization precision. For each pixel, the intensities of overlapping Gaussian spots were summed to determine the pixel intensity, which was then visualized using an appropriate colormap.

## 3D super-resolution imaging
To assess the feasibility of 3D SMLM in the SDC-OPR setup, we applied the single-molecule fitter for arbitrary PSF model method developed by the Ries lab, which enabled us to reconstruct 3D data from 2D

images without the need to add additional optics to the commercial SDC-OPR set-up[25]. Image stacks of beads (TetraSpeck™ Microspheres, 0.1 μm, ca. T7279) immobilized on a coverslip were acquired in a range of ±2000 nm with respect to the glass slide, using a z spacing of 15 nm and the same acquisition parameters as in Fig. 5. These videos were used to generate an experimental 3D PSF model using the 'calibrate3DsplinePSF' plugin on SMAP software[25]. Recorded videos of mitochondria were fitted using this experimental 3D model of the PSF. Results of z calibration are shown in Supplementary Fig. 10a.

## Clustering and quantitative-PAINT analysis
Clustering of structures of Fig. 4d (left panel) was done using DBSCAN of Picasso software. Regarding DBSCAN parameters we use a circle with radius epsilon ('eps') of 10.8 nm and minimum number of points ('minPts') of 15 localizations. The maximum distance within each cluster was measured to plot the histogram of Fig. 4d (right panel), that was later fitted with a multi-peak Gaussian with each peak centers multiples of the first

center: $f(x) = a_1 e^{-\left(\frac{x-b_1}{c_1}\right)^2} + a_2 e^{-\left(\frac{x-2b_1}{c_2}\right)^2} + a_3 e^{-\left(\frac{x-3b_1}{c_3}\right)^2}$.

Collagen-IV images were analyzed via quantitative-PAINT (qPAINT)[34–38] with a custom MATLAB (v.2022a) code that analyses the fluorescence time series of each detected cluster of localizations to estimate the number of proteins in each cluster. First, the images were clustered using DBSCAN with the following parameters: 'eps' set to 20 nm (to capture vesicle structures) and 'minPts' set to 10 localizations (in accordance to the imaging conditions). Localizations corresponding to the same cluster were grouped, and their time stamps (frame number) were used to reconstruct the sequence of dark times per cluster (i.e., continues frames without detected events). All the dark times per cluster were pooled together to generate a normalized cumulative histogram, which was then fitted with the following exponential function: $1 - e^{\frac{t}{\tau_d}}$, where $\tau_d$ represents the dark time per cluster. The inverse of the dark time was calculated for each cluster and stored as the qPAINT index of the cluster ($\xi$).

To estimate the number of proteins per cluster, a calibration was performed using the DNA-PAINT data of all measured FOVs. For each DNA-PAINT dataset, a histogram of qPAINT indices for small clusters (i.e., clusters with a maximum point distance of 55 nm) was generated. This histogram was fitted with a multi-peak Gaussian function, with peaks appearing at multiples of a qPAINT index of 0.0022 s$^{-1}$. This calibration value, corresponding to the qPAINT index for one single binding site, was used to estimate the number of proteins per cluster as the ratio of the qPAINT index of the cluster, and the calibration value. The histogram of Fig. 4e (left panel) shows the distribution of proteins per cluster for all analyzed FOVs.

## Localization precision and resolution
In this manuscript, the Cramér-Rao lower bound (CRLB) of the single-molecule fit is used to determine the localization precision values informed ($\sigma_{SMLM}$). The reported value is the average between $\sigma_x$ and $\sigma_y$, which are the mode value of the distribution of the localization precision in x and y, respectively.

We also provided the localization precision determined using closest neighbor analysis (NeNA)[20]. Finally, the resolution can be estimated as 2.35 $\sigma_{SMLM}$. In our example, the best theoretical resolution obtained was 5 nm, for the DNA origami sample measured using the SCD-OPR system at 4× magnification, in line with the super-resolved images obtained as docking pairs separated by 6 nm are resolvable.

## Statics and reproducibility
No statistical method was used to predetermine sample size, and no data were excluded from the analyses. No sample size calculation was performed. In general, sample sizes were kept as big as practically

possible with the described microscopy technique and type of illumination.

## Reporting summary

Further information on research design is available in the Nature Portfolio Reporting Summary linked to this article.

## Data availability

Single-molecule localization data generated in this study have been deposited in the Figshare repository, https://doi.org/10.6084/m9.figshare.28741373. Source data are provided with this paper.

## Code availability

Raw Image processing was performed using Picasso available via GitHub at https://github.com/jungmannlab/picasso and/or SMAP software available at https://github.com/jries/SMAP. qPAINT analysis code is accessible at https://github.com/Simoncelli-lab/qPAINT_pipeline.

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

## Acknowledgements

This research was funded by the Human Frontier Science Program Organization (HFSP) through a cross-disciplinary post-doctoral fellowship (LT0025/2023-C) to C.Z., the Biotechnology and Biological Sciences Research Council (BBSCR) through the London Interdisciplinary Doctoral Programme (BB/T008709/1) to M.D.J., the Engineering and Physical Sciences Research Council (EPSCR) to fund O.P.L.D. doctoral studies (EP/R513143/1 and EP/T517793/1), and the Royal Society through a Dorothy Hodgkin fellowship (DHF\R1\191019) to S.S. F.P.'s lab acknowledges support from the Medical Research Council (MRC) (MR/V001256/1 and M/Y012089/1 to F.P.) and from the BBSRC (BB/Y002075/1 to F.P. and R.W.W.). G.P.A. acknowledges support from the Swiss National Science Foundation (200021_184687) and through the National Center of Competence in Research Bio-Inspired Materials (NCCR, 51NF40_182881), the European Union Program HORIZON-Pathfinder-Open: 3D-BRICKS, grant Agreement 101099125. This work has also been supported by The Chan Zuckerberg Initiative "Multi-color single molecule tracking with lifetime imaging" (2023-321188) and BBSCR, BB/Y513064/1 to S.S. We thank Andrew Vaughan and Ki Hng of the LMCB light microscopy facility for their assistance with the Nikon CSU-W1 SoRA. We also thank Robert Tetley from Nikon for his assistance with NIS-Elements software for the deconvolution of the images.

## Author contributions

C.Z. and S.S. designed DNA origami samples. C.Z. conducted DNA origami, Nup96, microtubules, mitochondria, and *Drosophila* retina experiments, developed analysis software and analyzed the data. M.D.J. conducted preliminary DNA origami experiments. O.P.L.D. developed DNA-antibody conjugations. K.K., G.C., and G.P.A. fabricated the DNA origami structures. R.F.W. prepared *Drosophila* samples. C.Z., M.D.J., O.P.L.D., and S.S. conceived the study and designed the experiments. C.Z., O.P.L.D., W.R., F.P., and S.S. interpreted data and wrote the manuscript. All authors reviewed and approved the final manuscript.

## Competing interests

The authors declare no competing interests.
