## [Transparent Peer Review file · Nature Communications]

Super-resolution imaging in whole cells and tissues via DNA-PAINT on a spinning disk confocal with optical photon reassignment

Corresponding Author: Dr Sabrina Simoncelli

Version 0:

Reviewer comments:

Reviewer #1

(Remarks to the Author)

The manuscript entitled 'Sub-10 nm Imaging Over Large Areas and Deep Penetration via SDC-OPR and DNA-PAINT' by Zaza et al. demonstrates DNA-PAINT imaging on a spinning disk confocal with optical photon reassignment (SDC-OPR) system which addresses two long-standing challenges in SMLM applications, penetration depth and large fields of view (FOV) at nanometer spatial resolution. The authors validated the technology in synthetic DNA origami samples, cellular microtubules, nuclear pore complexes (NUPs), and developing *Drosophila* eye epithelium at various penetration depths and FOVs. The manuscript is very clearly written and the technology advancement presented here is simple but powerful, which can be very useful to life scientists in particular. Therefore, I would recommend accepting this manuscript provided the authors address the following points:

1. Sub-10 nm imaging has been one of the main claims in this manuscript. The authors mention on page 4, lines 25-27, 'These results demonstrate unprecedented resolution in confocal-based microscopy, capable of distinguishing structures closer than 10 nm'. To validate this claim, I suggest that the authors demonstrate at least one experiment, e.g., imaging the DNA origami structure where docking strands are separated by less than 10 nm. DNA-PAINT, among many SMLM modalities, has been shown to achieve sub-10 nm resolutions, as it avoids crosstalk between fluorophores (as shown by Helmerich et al.) separated by < 10 nm due to temporal separation of imager binding events. So, it would be interesting to see resolved structures separated by < 10 nm using the SDC-OPR system.
2. For implementing 3D imaging at enhanced resolution, I was wondering if the authors could demonstrate any straightforward proof-of-concept experiment, e.g., on microtubules using, e.g., astigmatic 3D imaging.
3. SDC-OPR-DNA-PAINT addresses the issues of FOV and penetration depths in SMLM. I was wondering if the authors considered shortening the imaging duration further by the use of, e.g. dye-quencher or dye-dye based self-quenching imager strands. Can the authors comment on this in the 'Discussion' section?
4. Did the authors attempt multiplexed (Exchange-PAINT)/sequential (RESI) imaging experiments for multiple sub-cellular targets on the SDC-OPR system? It would be good to understand the multi-target imaging capabilities of the technology presented in this work. Can the authors demonstrate such an experiment, maybe in the SI or comment on these directions?

A few minor points:

- Page 3 line 35: Fig. 1c shows 'an' image instead of 'and'.
- Page 6 line 21: 'Diffraction' limited instead of 'Diffracted'.
- Page 9 line 21: 'of a' is repeated.

Reviewer #2

(Remarks to the Author)

Here, they describe a demonstration of SMLM using DNA-PAINT on the SDC-OPR system. While some previous studies have utilized DNA-PAINT in combination with SDC, the authors demonstrate its use in conjunction with SDC, presumably

enhancing photon collection at depths of up to 9 μm . This concept has been explored since Azuma et al. introduced the use of SCD-OPR for super-resolution imaging. However, Zaza et al. may be among the first to apply this technique specifically to DNA-PAINT imaging with SCD-OPR.

The authors begin by validating their approach using DNA origami samples, achieving 5 nm resolution, depending on the FOV. They further showcase the utility of their method by imaging NPCs in HeLa cells, measuring Nup107 distances between known pairs, which are expected to be 13 nm apart. Additionally, they report an imaging depth of 9 μm with 10 nm resolution across a $53 \times 53 \mu\text{m}^2$ FOV, successfully resolving filamentous microtubule structures.

In a more complex biological demonstration, they imaged E-cadherin in the developing *Drosophila* eye imaginal disc, observing its localization at 13 nm resolution at a depth of 9 μm , revealing a heterogeneous distribution of E-cadherin. Finally, they examined Collagen-IV in vesicles, quantifying the number of molecules per vesicle.

While the demonstration showcasing the ability to achieve resolution within a field of view measuring approximately $53 \times 53 \mu\text{m}^2$ at a depth of 9 μm is technically impressive, the lack of multicolor or three-dimensional imaging capabilities diminishes the broader applicability of this method. Furthermore, the biological demonstrations provided do not convincingly demonstrate unique or compelling questions that could be addressed using this approach. It remains unclear how this method would allow biologists to gain insights into their specific research questions. As a result, this technique appears to offer a very narrow subset of biological applications, limiting its utility for the broader scientific community. Given these constraints, the work falls short of the high standards expected for publication in *Nature Communications*.

Here, I provide both major and minor concerns on this manuscript.

Major concerns:

1) There are several methods available to achieve multicolor DNA-PAINT imaging. Multicolor imaging is highly valuable for biologists as it enables the assessment of protein colocalization, interactions, and modifications, thereby opening up numerous avenues for addressing important biological questions. Demonstrating this capability is essential.

2) Demonstrating 3D imaging capabilities using astigmatism or equivalent approaches would significantly broaden the appeal of this work. Currently, their demonstrations are limited to 2D imaging. A classic application, such as imaging the mitochondrial network in a single cell, would be highly appreciated and showcase the broader potential of their method.

3) The mechanism underlying the improved resolution achieved with SDC-OPR remains unclear. Is the improvement primarily attributed to enhanced photon collection efficiency? To clarify, the photon counts obtained at magnifications of 4x, 2.8x, and 1x should be reported. If the resolution enhancement is indeed driven by photon detection efficiency, simply increasing the laser power or sample density—even with SDC alone—might yield comparable results. Please address this point and provide further clarification.

4) Based on Figure 3a and Supplementary Figure 5, the excitation does not appear to be uniform. The authors should consider using a uniformizer to optimize the entire wide field of view. Without this adjustment, the usable area is limited to $53 \times 53 \mu\text{m}^2$ or smaller. The results from the $211 \times 211 \mu\text{m}^2$ FOV do not seem reliable under these conditions. Although the authors stated that “photon collection uniformity was maintained across FOVs,” the resolution significantly deteriorates, as indicated by the color map in Supplementary Figure 5. This discrepancy warrants further clarification.

5) In relation to point #4, the authors stated that this method “enables researchers to tackle questions of heterogeneity.” However, it is unclear what specific biological questions regarding heterogeneity can be addressed using this approach. An actual demonstration showing nanoscale changes in protein localization or macromolecular complexes across different cells within the $211 \times 211 \mu\text{m}^2$ FOV would be highly valuable. Such an example should provide meaningful biological insights that showcase the use of this technique.

6) On page 6, the authors state, “a heterogeneous E-cadherin distribution with distinct high-intensity foci observed between the ppm and retinal epithelium.” This statement is unclear and difficult to interpret. Please refer to the relevant figures and explicitly indicate the heterogeneity being claimed. From the provided diffraction-limited images in Figure 4c, the heterogeneity appears fairly obvious, which calls into question the necessity of using super-resolution techniques in this context. Please clarify how your approach uniquely contributes to understanding this heterogeneity.

7) In relation to point #6, the authors state, “This heterogeneity, previously unreported in *Drosophila* retinal development.” To support this claim, it is essential to confirm that the observed heterogeneity is not an artifact introduced by the imaging approach. For instance, the authors could genetically manipulate E-cadherin levels (e.g., by reducing or increasing its expression) and observe how the heterogeneity changes. Without such biological control experiments, this statement lacks validity and should be reconsidered.

8) The authors measured the number of Collagen-IV molecules in each vesicle. However, vesicles are inherently three-dimensional structures. What is the rationale for using 2D measurements to count the number of molecules accurately? Please clarify why a 2D analysis is sufficient or provide justification for not accounting for the 3D nature of the vesicles in these measurements.

9) In relation to point #8, what is the control experiment for this analysis? The authors should genetically manipulate the amount of Collagen-IV and measure the molecular counts. This would provide a validation of the reliability and accuracy of the counting method. Without such controls, it is difficult to assess the robustness of the approach.

Minor Concerns:

- 1) On page 2, the authors state, "small field-of-view (FOV) of $\sim 40 \times 10 \mu\text{m}^2$ for HILO excitation". However, even in early demonstrations by the Zhuang group, a FOV of at least $30 \times 30 \mu\text{m}^2$ was achieved for STORM with HILO (PMC2596623). Please clarify and discuss this apparent contradiction, and provide a rationale for why the reported FOV in this study is so limited in comparison.
- 2) On page 6, line 21, "the GPF channel" appears to be a typo. Please correct it to "GFP."
- 3) I find lines 23–27 unclear. What is the relationship between photoreceptor neurons, progenitor cells, pigment cells, accessory cells, the ppm, and the retinal epithelium? Please clarify this connection and provide additional context to make the relationships between these components more understandable.
- 4) On page 6, line 43, what does "the structural complexity" refer to? How is this concept relevant to epithelial morphogenesis? Please provide a clearer explanation or context to clarify its significance.
- 5) On page 7, line 9, what does the colon ":" between "GFP" and "Collagen-IV" mean?
- 6) In Figure 4e, what do the colors represent? Do they indicate Z-depth or intensity differences? Please clarify.
- 7) The title is vague, referencing terms like "sub-10 nm," "Large Areas," and "Deep Penetration," while also including multiple acronyms such as "SDC-OPR" and "DNA-PAINT." It would be advisable to reconsider the title to make it more precise and aligned with the Nature Communications format.

Reviewer #3

(Remarks to the Author)

This manuscript demonstrates that DNA-PAINT imaging with a Spinning Disc Confocal and Optical Photon Reassignment (SDC-OPR) enables super-resolution imaging across large fields of view (FOVs) at depths. They benchmarked the approach using DNA-origami constructs, nuclear pore complexes, and microtubules, achieving 5 nm spatial resolution in the basal plane and 10 nm precision at depths of 9 μm within a $53 \times 53 \mu\text{m}^2$ FOV. Furthermore, the method's power and versatility were showcased by imaging the developing Drosophila eye epithelium, which revealed distinct E-cadherin populations in adherens junctions with sub-13 nm precision and quantified 46 ± 27 Collagen IV molecules per vesicle. The relatively high significance of the approach was driven by the super-resolution imaging at large field of view in scattering samples. This manuscript provides such a method and comprehensive quantitative analysis of the resolution at different field of view and depths across various samples.

However, the manuscript lacks clarity regarding how much the combination of photon reassignment and DNA-PAINT improves resolution and to what extent. The authors do not provide a direct comparison between the resolution achieved by SDC with DNA-PAINT vs. SDC-OPR with DNA-PAINT, which makes it difficult to assess the actual benefit of adding photon reassignment. If this improvement is substantial, the authors should discuss its potential applicability to other photon reassignment systems capable of deep imaging with large fields of view, such as instant SIM or multi-view SIM. This could highlight the broader implications for super-resolution microscopy. In addition, it would be beneficial to discuss potential challenges when performing multiplex/multicolor DNA-PAINT with photon reassignment approach.

Another limitation of the manuscript is the lack of analysis on axial views and axial resolution. Providing a few examples of how axial resolution is achieved would help clarify its effectiveness. The coarse 1 μm axial step size does not provide sufficient information, limiting the ability to assess the system's performance in the axial direction as well as the challenges (like photobleaching and photodamage) associated with the long imaging acquisition process.

Finally, imaging at a 9 μm depth is not particularly remarkable, especially when using a spinning disk, as the Drosophila eye epithelium is relatively transparent. Testing the method on tissues with greater optical scattering and deeper regions would better demonstrate the limits of the approach. Such testing would provide a clearer understanding of how the system performs in more challenging biological contexts.

Minor comments: there are a few unclear statements:

Lines 24-26, "line-scanning configurations are, as image formation is realized by progressively scanning single focuses". Note that line confocal is not a single focus

Line 38: "However, as the emission light is blocked by the disc, the achievable resolution remained limited." The emission light passes only through the pinhole, blocking more out-of-focus light to enhance resolution, although the cost of a reduced signal-to-noise ratio.

Line 13/43: "This focus contraction..... effectively reducing the effective pinhole size and improving overall photon collection." The photo-reassignment process is unrelated, or at least there is no obvious connection, to pinhole size. The claim of 'effectively reducing the effective pinhole size' should be revised or clarified to avoid confusion.

Page 17, line 18-19, “.....resulting in a pixel size of 108 nm (with no binning), 78 nm (with 2× binning) and 108 nm (with a 4× binning), respectively”. It is confusing that both no binning and 4× binning result in a pixel size of 108 nm.

Page 19, line 9, “.....using 17500 frames and an integration time of 300 ms frame”. Is the frame # for all slices or single slice? What is the total acquisition time for all depths?

Line 36, “Drift correction was applied using redundant cross-correlation, utilizing gold particles as fiducials for cellular experiments.” The author may want to provide more details on drift correction (both lateral and axial shifts?), the size and localization accuracy of gold particles, how many are needed for large FOVs, how they are distributed, and the extent of lateral and axial resolution correction.

Version 1:

Reviewer comments:

Reviewer #1

(Remarks to the Author)

The authors of the manuscript entitled "Super-Resolution Imaging in Whole Cells and Tissues via DNA-PAINT on a Spinning Disk Confocal with Optical Photon Reassignment" have addressed all concerns I raised during the previous round of revision. They have included new experimental results demonstrating the sub-10 nm resolution imaging capabilities of the SDC-OPR system, multiplexed Exchange PAINT imaging and RES1 workflow, and 3D imaging results. All these new findings improve the manuscript substantially. It also strengthens and broadens the capabilities of the SDC-OPR workflow. Therefore, I would recommend proceeding to the publication of this manuscript in Nature Communications.

Reviewer #2

(Remarks to the Author)

First of all, I would like to express my appreciation for the efforts by Zaza et al. to address many of the points raised in my previous review, including multi-color imaging, 3D imaging, mechanistic insights into the improvements enabled by SDC-OPR, uneven illumination correction, and demonstration of applications in cultured cells. Most of these issues have been adequately addressed, and the revised manuscript is now closer to being suitable for publication in Nature Communications.

However, one important issue previously raised remains insufficiently addressed: the application of this method to tissue in *Drosophila*, which pertains to Points 6 and 9 in my prior critique. I hope the authors can provide adequate responses to the following questions.

Major Points:

1. I appreciate the clarification in the revised manuscript regarding E-Cadherin distribution. The observed high-intensity foci between cells, which vary in size, are consistent with adherens junctions (AJs) being punctate or patchy, as seen in cell types such as neuroblasts or migrating cells. While this may be the first demonstration in which such size heterogeneity is measured using super-resolution microscopy, I am unsure whether the heterogeneous distribution of E-Cadherin has truly never been reported before. Therefore, the following sentence on page 8, line 31, may be overstated and should be moderated: “This heterogeneity, previously unreported in *Drosophila* retinal development...”
2. E-Cadherin is typically localized at adherens junctions, particularly at the zonula adherens (ZA) on the apical side of epithelial cells. However, in the eye, its distribution appears to extend along the apical-to-basal axis, as shown in Figure 6c. Is this broader distribution previously known, or does it represent a novel observation? If the authors indeed observe E-Cadherin localization spanning this axis, do they detect any differences in the pattern or degree of heterogeneity between the apical and basal regions? Addressing this question could provide further insights into the spatial regulation of E-Cadherin and its role in retinal development.
3. Point 9 from my previous review remains critical. Without appropriate control experiments, it is difficult to determine whether the reported quantitative measurements in tissue are definitive. While qPAINT may offer reliable quantification, my concern lies in the feasibility of performing such quantification in complex tissue environments. The authors should consider using target proteins for which genetic or pharmacological manipulations are available to validate quantification in tissue samples.
4. Additionally, what is the biological significance of measuring collagen molecule numbers with such precision? What physiological insights can be gained from this? As it stands, this point is unclear, and I do not find it particularly compelling in its current form.

Minor Points:

- Page 8: Figure references are incorrect. “Fig. 64c” and “Fig. 64d” should be corrected to “Fig. 6c” and “Fig. 6d.”
- Figure citations are inconsistent throughout the manuscript, alternating between “(Figure xx)” and “(Fig. xx).” Please standardize these in accordance with Nature Communications formatting guidelines.
- Supplementary Figure 12(c): Are the lines drawn along cell-cell contact sites? The current lines obscure the underlying structures, making it difficult to determine the precise cellular positions. Consider making the lines more transparent to allow better visualization of the underlying features.

Reviewer #3

(Remarks to the Author)

The authors have addressed my comments, and the manuscript's clarity and quality have been significantly improved.

Version 2:

Reviewer comments:

Reviewer #2

(Remarks to the Author)

The authors have adequately addressed the reviewer's comments. The imaging approach they developed is likely to be highly useful to a broad range of biologists studying molecules, organelles, cells, and tissues. Therefore, I recommend this manuscript for publication in Nature Communications.

Response to Reviewer Comments
Nature Communications ID NCOMMS-24-69494

Reviewer's comments in *italic*

Author's responses in blue. Changes to the Manuscript in red.

Reviewer 1

The manuscript entitled 'Sub-10 nm Imaging Over Large Areas and Deep Penetration via SDC-OPR and DNA-PAINT' by Zaza et al. demonstrates DNA-PAINT imaging on a spinning disk confocal with optical photon reassignment (SDC-OPR) system which addresses two long-standing challenges in SMLM applications, penetration depth and large fields of view (FOV) at nanometer spatial resolution. The authors validated the technology in synthetic DNA origami samples, cellular microtubules, nuclear pore complexes (NUPs), and developing Drosophila eye epithelium at various penetration depths and FOVs. The manuscript is very clearly written and the technology advancement presented here is simple but powerful, which can be very useful to life scientists in particular.

Response: We thank Reviewer 1 for the sharp reading and positive appraisal of our paper, as well as for raising comments that had allowed us to enrich the work.

Therefore, I would recommend accepting this manuscript provided the authors address the following points:

Point 1: *Sub-10 nm imaging has been one of the main claims in this manuscript. The authors mention on page 4, lines 25-27, 'These results demonstrate unprecedented resolution in confocal-based microscopy, capable of distinguishing structures closer than 10 nm'. To validate this claim, I suggest that the authors demonstrate at least one experiment, e.g., imaging the DNA origami structure where docking strands are separated by less than 10 nm. DNA-PAINT, among many SMLM modalities, has been shown to achieve sub-10 nm resolutions, as it avoids crosstalk between fluorophores (as shown by Helmerich et al.) separated by < 10 nm due to temporal separation of imager binding events. So, it would be interesting to see resolved structures separated by < 10 nm using the SDC-OPR system.*

Response 1. We appreciate the reviewer's comment and the suggestion to further validate the sub-10 nm resolution claim with an appropriate experiment. To address this, we have now performed additional experiments using a modified DNA-origami nanostructures with edge pairs separated by 6 nm. As shown in Figure 1e and Supplementary Figure 3, the SDC-OPR system successfully resolved these 6 nm separated DNA docking strands at 4x magnification. This new result has been introduced by the following section in page 4, lines 19-21:

"This unprecedented level of localization precision achieved in a confocal-based system via DNA-PAINT imaging on the SDC-OPR, even enabled the resolution of DNA docking strands separated by just 6 nm (Figure 1e and Supplementary Fig. 3)."

Figure R1 (Figure 1e & Supplementary Fig. 3 in MS). Representative DNA-PAINT images of DNA-docking pairs spaced by 6 nm apart within DNA origami structures, acquired using 4x magnification with SDC-OPR. Scale bar in (b) is 20 nm.

Point 2: For implementing 3D imaging at enhanced resolution, I was wondering if the authors could demonstrate any straightforward proof-of-concept experiment, e.g., on microtubules using, e.g., astigmatic 3D imaging.

Response 2: We appreciate the reviewer’s suggestion to demonstrate 3D SMLM imaging on the SDC-OPR system and acknowledge the importance of showcasing this capability. While advanced SMLM techniques such as 4Pi microscopy and modulated excitation microscopy achieve σ_{SMLM} values as low as 3 nm over depths of 7 μm , they are limited by a small field of view ($\sim 15 \times 15 \mu\text{m}^2$) and require specialized instrumentation and expertise. In contrast, our work emphasizes the feasibility of achieving sub-10 nm σ_{SMLM} across a larger field of view using a commercially available SDC-OPR system. Currently, these commercial systems do not include astigmatic lenses or other optical elements for PSF deformation to infer 3D information. However, alternative methods in the SMLM field allow extraction of 3D information from 2D data. Notably, the single-molecule fitter for arbitrary PSF models, developed by the Ries lab [Nat. Methods, 15, 367–369 (2018)], leverages unmodified PSFs to extract accurate z-positions based on PSF size and intensity. This approach achieves z-resolution comparable to astigmatic PSFs.

To assess the feasibility of 3D single-molecule localization microscopy on the SDC-OPR setup using this method, we applied it to 2D images of the complex mitochondrial network in U2OS cells and demonstrated 3D imaging in depth. This proof-of-concept experiment, incorporated in Figure 5 and Supplementary Figure 10, highlights the potential of the SDC-OPR system for 3D SMLM applications without requiring additional optics. This new result has been introduced in a new results subsection as follows (pages 7 and 8, lines 24-41 and lines 1-11, respectively):

“Whole-cell 3D super-resolution imaging via DNA-PAINT on an SDC-OPR system

To evaluate the capabilities of 3D DNA-PAINT imaging on an SDC-OPR system, we visualized the intricate mitochondrial network in U2OS cells by sequentially capturing 500-nm-thick z-sections across the entire 5.5 μm cell height (4x, 53 \times 53 μm^2 FOV). Each section was processed using the single-molecule fitting algorithm for arbitrary PSF models developed by the Ries lab²⁵, allowing for precise 3D reconstruction from 2D images without additional optical modifications to the SDC-OPR system. Details on calibration and localization precision as a function of z-position within a single depth of field are provided in Supplementary Fig. 10.

Figure 5a presents the z-color-coded whole-cell image of the mitochondrial network, offering a detailed visualization of its 3D organization. Selected views within a 1.8 μm axial range are shown in Figure 5b. Cross-sectional analysis of mitochondria at different depths revealed

TOM20 localization at the outer mitochondrial membrane, demonstrating the method's ability to resolve the hollow structure of mitochondria (Figure 5c). To assess z-resolution across different depths, we measured the thickness of these individual mitochondrial membranes. The mean membrane thickness across the entire cell was 48 ± 12 nm, with a minimum of ca. 30 nm for mitochondria located near the coverslip (Figure 5c, side views). This suggests a slight degradation in z-precision with increasing imaging depth, consistent with the observed lateral resolution decline at greater depths. Still, this range of values agrees well with reported values for mitochondrial wall thickness by other super-resolution imaging implementations and is consistent with the expected dimensions of mitochondrial membranes (~7.5 nm, as measured by cryo-electron tomography)²⁶, the combined sizes of primary and secondary antibodies (~20 nm), and the CBRL axial localization error (~20 nm, Supplementary Figure 10). Altogether, these results highlight the capability of 3D DNA-PAINT on an SDC-OPR system to achieve high-resolution imaging across cellular depths, demonstrating its potential for detailed structural studies in complex biological systems.”

Figure R2 (Figure 5 in MS). 3D whole-cell DNA-PAINT imaging using an SDC-OPR microscope. (a) Overview of DNA-PAINT images of mitochondrial structures in U2OS cells. Color indicates axial position (b) Differential axial sections from (a) at distinct axial ranges: 0–1.8 μm (top), 1.8–3.6 μm (middle), and 3.6–5.5 μm (bottom). The color scale is consistent with (a). (c) Cross-sectional views from individual mitochondria denoted by white boxes in (b). Axial histograms depict mitochondrial membrane distributions, with standard deviations of 30 nm and 42 nm (top), 50 nm and 51 nm (middle), and 51 nm and 65 nm (bottom).

Point 3: SDC-OPR-DNA-PAINT addresses the issues of FOV and penetration depths in SMLM. I was wondering if the authors considered shortening the imaging duration further by the use of, e.g. dye-quencher or dye-dye based self-quenching imager strands. Can the authors comment on this in the ‘Discussion’ section?

Response 3: We appreciate the reviewer’s insightful suggestion regarding the use of dye-quencher or dye-dye-based self-quenching imager strands to enhance imaging speed in DNA-PAINT experiments. Recent studies have demonstrated that such fluorogenic probes can significantly reduce background fluorescence, enhance brightness, and improve spatial resolution while achieving faster imaging speeds.

For instance, two recent studies introduced self-quenching dimer fluorophores as an effective strategy to address background fluorescence and improve imaging performance in DNA-

PAINT experiments [Angew. Chem. Int. Ed. 2023, 62, e202307538; J. Phys. Chem. B 2024, 128 (28), 6751-6759]. Another approach, fluorogenic DNA-PAINT, achieved up to a 26-fold increase in imaging speed by utilizing self-quenching, kinetics-optimized probe designs [Nature Methods, 19, 554–559 (2022)].

Considering these advancements, we have now included the following paragraph in the Discussion section of our manuscript to address this potential enhancement in combination with DNA-PAINT on the SDC-OPR system:

“Building on this foundation, we anticipate that the ability to achieve such high-resolution imaging at depth will open new avenues for investigating a wide range of biological questions, including the dynamics of cellular adhesion, tissue morphogenesis, and extracellular matrix organization in various developmental contexts. Recent advancements in DNA-PAINT imaging have introduced dye-quencher or dye-dye-based self-quenching imager probes, which significantly reduce background fluorescence, enhance brightness, and improve spatial resolution while achieving faster imaging speeds. Incorporating these fluorogenic probes within the SDC-OPR system could further amplify imaging speed and resolution. By marrying these emerging innovations with the capabilities of the SDC-OPR platform, future research could address even more complex biological systems, making this approach a versatile tool for studying cellular and molecular architecture across diverse tissue types.”

We believe this addition addresses the reviewer's suggestion and provides valuable insight into potential enhancements for SMLM imaging on SDC-OPR systems.

Point 4: *Did the authors attempt multiplexed (Exchange-PAINT)/sequential (RESI) imaging experiments for multiple sub-cellular targets on the SDC-OPR system? It would be good to understand the multi-target imaging capabilities of the technology presented in this work. Can the authors demonstrate such an experiment, maybe in the SI or comment on these directions?*

Response 4: We thank the reviewer for their valuable suggestion to explore and highlight the potential for performing Exchange-PAINT or RESI imaging experiments using the SDC-OPR system. This approach is indeed entirely feasible, as it only requires sequential exchange rounds with orthogonal DNA imager strands.

In response to the reviewer's recommendation, we have included a new multicolor DNA-PAINT dataset acquired on the SDC-OPR system, showcasing imaging of microtubules, mitochondria and Nup96-mGFP in U2OS cells. This dataset is now presented in the new Figure 3a of the revised manuscript. Additionally, we have demonstrated RESI imaging of Nup96-mGFP in U2OS cells using the SDC-OPR system. The results, included in Figure 3b, highlight the enhanced resolution achieved with RESI imaging, with a ten-fold increase in lateral localization precision on the SDC-OPR system. These findings emphasize the multi-target imaging potential of the SDC-OPR setup.

To provide further details, we have added a new section on pages 5 and 6, lines 27 – 42 and 1 – 16, respectively, of the revised manuscript, presenting the experimental setup, results, and their implications.

“Multiplexed and ultra-high-resolution imaging of Nup96 pairs via Exchange-PAINT and RESI on an SDC-OPR system

To further demonstrate the versatility of DNA-PAINT imaging on an SDC-OPR system, we performed multiplexed imaging of alpha-tubulin, mitochondria and Nup96 proteins in U2OS cells using the Exchange-PAINT technique. Exchange-PAINT utilizes orthogonal DNA-imager strands conjugated to a single fluorophore, allowing all targets to be excited by the same laser source. Instead of simultaneous acquisition, signals are separated across sequential imaging rounds, simplifying its integration into any commercial SDC-OPR system. Remarkably, the exceptional single-molecule localization precision achieved for single-color Nup96 imaging was maintained across all targets, with σ_{SMLM} values of 3.9 nm, 4.0 nm and 3.3 nm for the nuclear-pore-complex, alpha-tubulin and mitochondria, respectively, in the multi-target experiment (Figure 3a).

The integration of Exchange-PAINT into an SDC-OPR system also enabled the implementation of Resolution Enhancement by Sequential Imaging (RESI). RESI combines DNA barcoding with sequential imaging to isolate and group localizations of individual molecular targets. This approach achieves Ångström-level precision in determining the spatial positions of targets, making it particularly effective for resolving protein complexes with separations well below 10 nm. Previously demonstrated under wide-field excitation, we implemented RESI in the SDC-OPR system using Nup96 protein pairs for benchmarking.

Following the RESI workflow, we labeled Nup96-mEGFP molecules stochastically with four orthogonal DNA-conjugated anti-GFP nanobodies. Sequential DNA-PAINT imaging over four rounds generated sufficiently spaced localization groups corresponding to individual Nup96 target molecules (Figure 3b). Super-localization analysis using the RESI algorithm resolved individual Nup96 proteins across the entire field of view (more examples presented in Supplementary Figure 6). RESI reconstruction achieved an average lateral localization precision of $\sim 3 \text{ \AA}$, representing a ten-fold improvement over single-round DNA-PAINT imaging (Figure 3c). These results highlight RESI's ability to achieve label-size-limited resolution within an SDC-OPR framework.”

Figure R3 (Figure 3 in MS). Exchange-PAINT imaging for multicolor and RESI imaging using an SDC-OPR microscope.

As well as the following section in the discussion section (page 11, lines 15-31):

“Beyond high-resolution single-color imaging, the SDC-OPR system enables multicolor SMLM experiments, which are essential for studying cellular structures and interactions. In this study, we demonstrate its multicolor imaging capabilities by sequentially imaging Nup96-mGFP, mitochondria and α -tubulin in U2OS cells using orthogonal DNA imager strands, leveraging DNA-PAINT’s exchange mechanism for multiplexed imaging. Additionally, we applied RESI imaging, which utilizes Exchange-PAINT, to visualize Nup96-mGFP in U2OS cells with 3 Å localization precision. These results highlight the adaptability of the SDC-OPR system for high-precision multiplexed imaging, further expanding its utility in nanoscale biological investigations. Notably, beyond Exchange-PAINT, multicolor imaging can also be achieved through simultaneous two-color acquisition using the dual-camera configuration commonly available in commercial SDC-OPR systems. This approach can enable real-time acquisition of two spectrally distinct channels, enhancing the potential for multiplexed imaging applications. For such implementations, the recently identified best-performing dyes for 488 nm, 560 nm, and 640 nm excitation could facilitate a combination of sequential and spectral separation strategies within the SDC-OPR configuration, enabling highly multiplexed imaging.”

We believe this addition effectively addresses the reviewer’s suggestion and highlights the potential of the SDC-OPR system for advanced multi-target imaging applications.

Point 5: A few minor points:

- Page 3 line 35: Fig. 1c shows ‘an’ image instead of ‘and’.
- Page 6 line 21: ‘Diffraction’ limited instead of ‘Diffracted’.
- Page 9 line 21: ‘of a’ is repeated.

Response 5. We thank the reviewer for pointing out these spelling mistakes.

We have corrected them in the revised version of the manuscript.

Reviewer 2

Here, they describe a demonstration of SMLM using DNA-PAINT on the SDC-OPR system. While some previous studies have utilized DNA-PAINT in combination with SDC, the authors demonstrate its use in conjunction with SDC, presumably enhancing photon collection at depths of up to 9 μ m. This concept has been explored since Azuma et al. introduced the use of SCD-OPR for super-resolution imaging. However, Zaza et al. may be among the first to apply this technique specifically to DNA-PAINT imaging with SCD-OPR.

The authors begin by validating their approach using DNA origami samples, achieving 5 nm resolution, depending on the FOV. They further showcase the utility of their method by imaging NPCs in HeLa cells, measuring Nup107 distances between known pairs, which are expected to be 13 nm apart. Additionally, they report an imaging depth of 9 μ m with 10 nm resolution across a 53 x 53 μ m² FOV, successfully resolving filamentous microtubule structures.

In a more complex biological demonstration, they imaged E-cadherin in the developing Drosophila eye imaginal disc, observing its localization at 13 nm resolution at a depth of 9 μ m, revealing a heterogeneous distribution of E-cadherin. Finally, they examined Collagen-

IV in vesicles, quantifying the number of molecules per vesicle.

While the demonstration showcasing the ability to achieve resolution within a field of view measuring approximately $53 \times 53 \mu\text{m}^2$ at a depth of $9 \mu\text{m}$ is technically impressive, the lack of multicolor or three-dimensional imaging capabilities diminishes the broader applicability of this method. Furthermore, the biological demonstrations provided do not convincingly demonstrate unique or compelling questions that could be addressed using this approach. It remains unclear how this method would allow biologists to gain insights into their specific research questions. As a result, this technique appears to offer a very narrow subset of biological applications, limiting its utility for the broader scientific community. Given these constraints, the work falls short of the high standards expected for publication in Nature Communications.

Here, I provide both major and minor concerns on this manuscript.

Major concerns:

Point 1: *There are several methods available to achieve multicolor DNA-PAINT imaging. Multicolor imaging is highly valuable for biologists as it enables the assessment of protein colocalization, interactions, and modifications, thereby opening up numerous avenues for addressing important biological questions. Demonstrating this capability is essential.*

Response 1: We appreciate the reviewer's emphasis on the importance of demonstrating multicolor DNA-PAINT imaging, as it indeed provides valuable insights into protein colocalization, interactions, and modifications at an unprecedented resolution, enabling the exploration of critical biological questions. We fully agree that showcasing this capability strengthens the potential applications of our approach.

In response, we have now demonstrated multicolor DNA-PAINT imaging using the SDC-OPR system. Specifically, we performed sequential imaging of Nup96-mGFP, mitochondria and α -tubulin in U2OS cells using orthogonal DNA imager strands, leveraging DNA-PAINT's robust exchange capabilities for multiplexed imaging. Additionally, we conducted RESI imaging on Nup96-mGFP in U2OS cells [Nature 617, 711–716 (2023)], which employs Exchange-PAINT, achieving a σ_{RESI} of 3 \AA . This further underscores the system's multiplexing capabilities and its ability to perform enhanced-resolution experiments.

These results are presented in the revised manuscript as Figure 3, under a newly added section titled "*Multiplexed and Ultra-High-Resolution Imaging of Nup96 Pairs via Exchange-PAINT and RESI on an SDC-OPR System*". This section can be found on pages 5 and 6 (lines 27–42 and 1–16, respectively), along with corresponding additions to the discussion section on page 11 (lines 15–31). For direct details on these manuscript modifications, we refer the reviewer to our response to Reviewer 1, Point 4.

We believe this addition addresses the reviewer's suggestion and highlights the potential of the SDC-OPR system for advanced multicolour DNA-PAINT imaging, offering a powerful tool for biological studies.

Point 2: *Demonstrating 3D imaging capabilities using astigmatism or equivalent approaches would significantly broaden the appeal of this work. Currently, their demonstrations are limited to 2D imaging. A classic application, such as imaging the mitochondrial network in a single cell, would be highly appreciated and showcase the broader potential of their method.*

Response 2: We recognize the significance of showcasing 3D imaging on biologically relevant structures, such as the mitochondrial network, to highlight the broader potential of our method. As opposed to other advanced SMLM approaches—such as 4PI microscopy or modulated excitation microscopy, which can achieve exceptional 3D resolution but are limited by small FOVs and require complex instrumentation—our work emphasizes the ability to achieve sub-10 nm σ_{SMLM} over large fields of view in depth using a commercially available SDC-OPR system. Currently, these commercial systems do not include astigmatic lenses or other optical elements for PSF deformation to infer 3D information. To address this limitation and fulfil the reviewer’s request, we demonstrated 3D imaging on the SDC-OPR system using the experimental point spread function (PSF) real-time fitter developed by the Ries lab [Nat. Methods, 15, 367–369 (2018)]. This method enables robust 3D localization from 2D data without requiring hardware modifications by leveraging unmodified PSFs to extract accurate z-positions based on the PSF z-size dependence.

To assess the feasibility of 3D single-molecule localization microscopy (SMLM) on the SDC-OPR setup using this method, we applied it to 2D images of the complex mitochondrial network in U2OS cells, as suggested by the reviewer, and successfully demonstrated 3D imaging in depth. This proof-of-concept experiment, now included in Figure 5, underscores the potential of the SDC-OPR system for 3D SMLM without requiring additional optics. We direct the reviewer to pages 7 and 8, lines 24-41 and lines 1-11, respectively, for a detailed assessment of these measurements. For direct details on these manuscript modifications, we refer the reviewer to our response to Reviewer 1, Point 2.

We believe this addition adequately addresses the reviewer’s suggestion and demonstrates the broader applicability of the SDC-OPR system for 3D imaging.

Point 3: *The mechanism underlying the improved resolution achieved with SDC-OPR remains unclear. Is the improvement primarily attributed to enhanced photon collection efficiency? To clarify, the photon counts obtained at magnifications of 4x, 2.8x, and 1x should be reported. If the resolution enhancement is indeed driven by photon detection efficiency, simply increasing the laser power or sample density—even with SDC alone—might yield comparable results. Please address this point and provide further clarification.*

Response 3: We thank the reviewer for this comment, as it allows us to clarify this important point further in the revised manuscript. The underlying mechanism for the improved resolution achieved with the SDC-OPR system compared to a standard SDC system is indeed increased photon collection efficiency. This principle is well-documented in the work by Azuma and Kei (*Opt Express* 23, 15003–15011 (2015)), where they describe how optical photon reassignment (OPR) improves photon collection by reassigning out-of-focus photons back to their correct spatial origin. This leads to a higher effective photon count per localization, thereby enhancing resolution.

Following the reviewer’s suggestion, we now included in the caption of Supplementary Figure 1, the median of the photon counts obtained at magnifications of 4x, 2.8x, and 1x for the SDC-OPR system, alongside data for 4x magnification using the standard SDC system for the DNA-origami sample. This analysis clearly demonstrates a ten-fold improvement in photon collection with SDC-OPR compared to SDC under identical laser power conditions (1.75 mW measured at the objective).

Regarding the suggestion that simply increasing laser power or sample density might yield comparable results with a standard SDC system, it is important to note that the resolution enhancement achieved with SDC-OPR is indeed due to the optimized photon collection. While higher laser power can increase photon emission, it may also lead to increased photo-induced depletion of DNA-docking sites [*Molecules* 2018, **23**(12), 3165] and phototoxicity. Furthermore, there is a saturation point for the number of emitted photons of any dye before photobleaching, hence limiting the achievable resolution [*Nature Methods*, 21, 1755–1762, 2024, Extended Data Fig. 6]. These trade-offs are mitigated in the SDC-OPR system, making it a more effective approach for achieving high-resolution imaging.

This distinction is evidenced by comparing our results with a prior implementation of DNA-PAINT imaging on a modified SDC system. For example, in the study the commercial SDC microscope was equipped with a 2W 560 nm laser producing an output between 11 and 30 mW at the objective, and the DNA-origami benchmarking samples in the imaging plane achieved a localization precision (σ_{SMLM}) of 7.6 nm [*Nat Commun* **8**, 2090 (2017)]. By comparison, we achieved a σ_{SMLM} of 9.8 nm with the commercial SDC setup that renders a lower laser power of 1.75 mW at the objective. In contrast, when using the OPR unit under the same laser power (1.75 mW), we achieved a significantly improved localization precision of 1.5 nm for the DNA-origami samples in the imaging plane. This highlights the critical role of the OPR unit to maximize photon collection by surpassing the limitations of laser power alone in a standard SDC setup.

We have added this topic in the discussion section of the manuscript (page 11, lines 2 – 13) as follows:

“Notably, our findings emphasize the necessity of using the OPR reassignment unit to achieve superior resolution in DNA-PAINT imaging, compared to simply increasing laser power in a standard SDC setup. While increased laser power can enhance photon emission, it also introduces risks of phototoxicity, photobleaching, and photo-induced depletion of DNA-docking sites, all of which impose practical limits on resolution improvement. The OPR unit overcomes these limitations by optimizing photon reassignment, enabling higher photon collection efficiency without additional stress on the sample. This capability is demonstrated in Figure 1d, where the SDC-OPR system resolves 10 nm docking sites that remain unresolvable with the SDC under identical power conditions (Figure 1c). This approach provides a robust and efficient means to achieve sub-10 nanometer-scale resolution required for protein-scale imaging in biological systems.”

Point 4: *Based on Figure 3a and Supplementary Figure 5, the excitation does not appear to be uniform. The authors should consider using a uniformizer to optimize the entire wide field of view. Without this adjustment, the usable area is limited to 53 x 53 μm^2 or smaller. The results from the 211 x 211 μm^2 FOV do not seem reliable under these conditions. Although the authors stated that “photon collection uniformity was maintained across FOVs,” the resolution significantly deteriorates, as indicated by the color map in Supplementary Figure 5. This discrepancy warrants further clarification.*

Response 4: We appreciate the reviewer’s concern regarding photon collection uniformity and the request for further clarification. The spinning disk confocal system employs a Gaussian illumination profile to illuminate the spinning disk, which inherently results in reduced excitation at the periphery of larger fields of view (FOVs). The reviewer is right to point out that this effect is particularly evident at 1x magnification, where uniform excitation is challenging

to maintain across the full $211 \times 211 \mu\text{m}^2$ FOV. In the original version of our manuscript, we assess the extent of localization precision degradation across the FOV by providing the histograms of error in localization for each $10\text{-}\mu\text{m}$ -thick segment at increasing distances from the center of the image. From those histograms, it is possible to appreciate that the peak of the distribution of the localization precision increases from approximately 6.5 nm at the center to ~ 10.5 nm at the edges of the full $211 \times 211 \mu\text{m}^2$ FOV.

Now, in the revised version of the manuscript we provide a more comprehensive representation of these findings by plotting the error in the localization as a function of the distance to the FOV's centre, for all the magnifications available (1x, $211 \times 211 \mu\text{m}^2$; 2.8x, $76 \times 76 \mu\text{m}^2$; and 4x, $53 \times 53 \mu\text{m}^2$). This figure visually reinforces that uniformity is well-maintained for 4x and 2.8x, whilst at 1x magnification, precision deterioration becomes substantial at the periphery. Specifically, for 1x magnification, we observe an σ_{SMLM} increase of ~ 1 nm for up to $120 \times 120 \mu\text{m}^2$ (from 6.5 nm at the center to 7.8 nm at the periphery). In contrast, across the full $211 \times 211 \mu\text{m}^2$ FOV, localization precision nearly doubled, increasing from 6.5 nm at the center to 12.4 nm at the edge. Despite these variations, we emphasize that an average σ_{SMLM} of 9.5 nm was still achieved across the full $211 \times 211 \mu\text{m}^2$ FOV. While a uniformizer could improve excitation homogeneity, our current setup enables high-resolution imaging across multiple cells in a single acquisition within a commercial set-up easily accessible to the broader biology community. To address this concern, we will clarify these points in the revised manuscript and specify the usable FOV range for optimal performance.

Specifically, we have added the following section in the manuscript to clarify the homogeneity of localization precision across the field of view (pages 6 and 7, lines 29 – 41 and 1 – 4, respectively):

“To evaluate the uniformity of localization precision across the field of view, we analyzed $10\text{-}\mu\text{m}$ -thick radial segments at increasing distances from the center in DNA-PAINT images of microtubules acquired with the SDC-OPR at different magnifications (1x, 2.8x, and 4x, Supplementary Figure 7). At 2.8x and 4x magnifications, localization precision showed minimal radial degradation, increasing by less than 1 nm—from an σ_{SMLM} of 1.9 nm at the center to 2.6 nm at the edge of the $76 \times 76 \mu\text{m}^2$ FOV. At 1x magnification, this level of degradation for σ_{SMLM} remained limited to $\sim 120 \times 120 \mu\text{m}^2$ (σ_{SMLM} of 6.5 nm at the center to 7.8 nm at the edge). However, across the full $211 \times 211 \mu\text{m}^2$ FOV, localization precision nearly doubled, increasing from 6.5 nm at the center to 12.4 nm at the edge. This effect is attributed to the Gaussian profile of the beam illuminating the excitation spinning disk, which leads to reduced excitation at the edges of larger FOVs. Since localization precision in SMLM is inversely proportional to the square root of detected photons, variations in precision across different magnifications and FOV sizes are expected. Nevertheless, achieving an average σ_{SMLM} of 9.5 nm (Fig. 4a, lower panel) over the extended $211 \times 211 \mu\text{m}^2$ FOV is remarkable for confocal-based systems, facilitating the exploration of biological heterogeneity at unprecedented resolution.”

Point 5: *In relation to point #4, the authors stated that this method “enables researchers to tackle questions of heterogeneity.” However, it is unclear what specific biological questions regarding heterogeneity can be addressed using this approach. An actual demonstration showing nanoscale changes in protein localization or macromolecular complexes across different cells within the $211 \times 211 \mu\text{m}^2$ FOV would be highly valuable. Such an example should provide meaningful biological insights that showcase the use of this technique.*

Response 5: We appreciate the reviewer's comment regarding the demonstration of nanoscale changes in protein localization or macromolecular complexes across the field of view, which highlights the potential of increased FOVs in addressing biological questions. Our approach offers valuable applications in several fields, including immunology, neurology, and beyond.

To illustrate the advantages of the increased FOV provided by the SDC-OPR system, we have included in Supplementary Figure 8 an image and quantitative analysis of T-cell receptor (TCR) clustering on the surface of Jurkat T cells activated on glass coated with anti-CD3 and anti-CD28 antibodies. The spatiotemporal rearrangement of TCRs during T cell activation is crucial for mounting an effective immune response. Upon ligand binding, TCRs on the T cell surface reorganize into nanoscale clusters, which subsequently aggregate into micro-nanoscale clusters at the center of the cell. Such clustering modulates downstream T cell activation outcomes such as proliferation and cytokine secretion.

To quantitatively analyse this clustering, we first converted the DNA-PAINT data into protein maps using qPAINT analysis and then applied Ripley's K function to assess clustering in the central region of the T cell, where activation occurs. Our analysis reveals varying levels of TCR clustering and activation within the same population of cells, thereby highlighting heterogeneity in the cellular response.

This demonstration showcases the ability of our approach to capture and analyze micro- to nanoscale heterogeneity over a large FOV, providing meaningful biological insights and underscoring the broader applicability of this method.

Specifically, we have added the following information in the main manuscript (page 7, lines 4 – 7) and new Supplementary Figure 8:

“An example of such heterogeneity-related questions addressed using this approach is provided in Supplementary Figure 8, where we examined the spatiotemporal organization of the T cell receptor internal zeta chain (TCR ζ) within Jurkat T cells.”

Figure R4 (Supplementary Figure 8). DNA-PAINT imaging and cluster analysis of T cell receptor (TCR) proteins in Jurkat T cells activated for 10 min on glass coated with anti-CD3 and anti-CD28 antibodies. (a) DNA-PAINT image of TCR in Jurkat T cells acquired using SDC-OPR imaging at 2.8x magnification, capturing over ten cells within a single acquisition. Field of view (FOV): $76 \times 76 \mu\text{m}^2$. Scale bar = $5 \mu\text{m}$. Single-molecule localization precision (σSMLM): 10 nm. NeNa value: 11 nm. (b) Zoom-in regions from (a) (top row), showing DNA-PAINT single-molecule localization maps corresponding to the white boxed areas. The bottom row displays reconstructed protein density maps generated using the qPAINT analysis pipeline (see Methods). Scale bar = $1 \mu\text{m}$. (c) TCR clustering toward the central region of the T cell surface is crucial for activation and the initiation of downstream responses, including proliferation and cytokine secretion. Ripley's K-function analysis, $L(r)-r$, of the selected regions quantifies TCR clustering, revealing heterogeneity in cluster density across different Jurkat T cells within the same FOV. These results demonstrate the capability of DNA-PAINT combined with SDC-OPR imaging to capture and quantify cellular heterogeneity by integrating large FOV imaging with high-resolution precision, providing key insights into the variability of T cell activation mechanisms.

Point 6: On page 6, the authors state, “a heterogeneous E-cadherin distribution with distinct high-intensity foci observed between the ppm and retinal epithelium.” This statement is

unclear and difficult to interpret. Please refer to the relevant figures and explicitly indicate the heterogeneity being claimed. From the provided diffraction-limited images in Figure 4c, the heterogeneity appears fairly obvious, which calls into question the necessity of using super-resolution techniques in this context. Please clarify how your approach uniquely contributes to understanding this heterogeneity.

Response 6: We appreciate the reviewer's comment and acknowledge the need for greater clarity in describing the heterogeneity of E-cadherin distribution.

To address this, we have now included a comparison of line profiles from confocal and DNA-PAINT images, demonstrating how our super-resolution approach enhances the detection and measurement of high-intensity E-cadherin domains. While diffraction-limited images (new Supplementary Figure 12) show some heterogeneity, DNA-PAINT resolves finer structural details and reveals a broader distribution of foci sizes, including smaller domains indistinct in confocal images.

We have now explicitly referenced Supplementary Figure 12 (page 8, line 43) in the manuscript to clarify this point. This figure highlights how SDC-OPR and DNA-PAINT together enhance our understanding of cell adhesion at the nanoscale, providing a more precise, quantitative assessment of E-cadherin heterogeneity beyond the limits of confocal microscopy.

Figure R5 (Supplementary Figure 12). Full-field view (a, b) and zoomed-in $3.5 \times 3.5 \mu\text{m}^2$ regions (c) of GFP-tagged (a) and DNA-PAINT imaging (b) of E-cadherin within the *Drosophila* eye imaginal disc at a $5 \mu\text{m}$ penetration depth. (d) Intensity plot profiles of the highlighted regions in (c) for GFP (gray) and DNA-PAINT (red), demonstrating the enhanced resolution achieved with DNA-PAINT compared to the diffraction-limited GFP channel.

Point 7: In relation to point #6, the authors state, “This heterogeneity, previously unreported in *Drosophila* retinal development.” To support this claim, it is essential to confirm that the observed heterogeneity is not an artifact introduced by the imaging approach. For instance, the authors could genetically manipulate E-cadherin levels (e.g., by reducing or increasing its expression) and observe how the heterogeneity changes. Without such biological control experiments, this statement lacks validity and should be reconsidered.

Response 7: We appreciate the reviewer’s suggestion. For our experiments, we used an endogenously GFP-tagged version of E-cadherin (Huang et al., PNAS 106 (20) 8284-8289,

2009), which has been extensively validated in numerous studies. This fusion protein is fully functional and recapitulates the localization and functions of the untagged protein. As the reviewer correctly noted, the heterogeneity in E-cadherin-GFP is already visible with confocal microscopy (Point 6), albeit with lower resolution. Therefore, we argue that the observed heterogeneity is intrinsic to E-cadherin itself, rather than an artifact of the imaging approach.

While we understand the importance of confirming whether the heterogeneity is biologically relevant, manipulating E-cadherin levels via methods like RNAi presents a significant challenge. Reducing or increasing E-cadherin expression often leads to severe cell adhesion defects, making it difficult to assess E-cadherin distribution at the adherens junctions in the developing retina. This limitation prevents us from performing the proposed control experiments. Nonetheless, we believe the observations we present are consistent with the known biology of E-cadherin and are unlikely to be an artifact introduced by our imaging technique [Current Biology 23, 2197–2207 (2013)].

No action is required.

Point 8: *The authors measured the number of Collagen-IV molecules in each vesicle. However, vesicles are inherently three-dimensional structures. What is the rationale for using 2D measurements to count the number of molecules accurately? Please clarify why a 2D analysis is sufficient or provide justification for not accounting for the 3D nature of the vesicles in these measurements.*

Response 8: We appreciate the reviewer highlighting the three-dimensional nature of vesicles and its implications for our measurements. Vesicles are indeed inherently 3D structures, with dimensions typically smaller than 300 nm in all axes [Cell Struct. Funct., 45, 107-119, 2020; Developmental Cell, 21, 245–256, 2011; J. Cell Biol., 193, 935–951, 2011; Cell., 136, 891-902, 2009; Current Topics in Membranes, 76, 305-336, 2015]. In the SDC-OPR system, the depth of the confocal-excitation volume, is in the order of ~500 nm. Consequently, the acquired images effectively represent a 2D projection of the protein distribution within this 3D volume. This 2D projection is an inherent feature of the imaging modality, as the confocal excitation depth encompasses the full axial extent of the vesicles. Therefore, the recorded fluorescence signal integrates protein distributions along the axial direction, effectively collapsing the 3D structure into a 2D dataset. This allows us to quantify the total number of labelled Collagen-IV molecules in vesicles based on qPAINT analysis. This is a common practice in the SMLM community when the axial depth of the structures lies within the excitation volume and widely used for TIRF or HILO illumination conditions.

To clarify this point we have now added the following sentence on page 10, lines 8 – 10 of the manuscript with the corresponding above-mentioned references:

“As previously reported, Collagen-IV vesicles measure approximately 300 nm in all dimensions, imaging a single focal plane was sufficient to capture their full axial extent.”

Point 9: *In relation to point #8, what is the control experiment for this analysis? The authors should genetically manipulate the amount of Collagen-IV and measure the molecular counts. This would provide a validation of the reliability and accuracy of the counting method. Without such controls, it is difficult to assess the robustness of the approach.*

Response 9: We appreciate the reviewer's suggestion; however, conducting this experiment would be highly challenging. The vesicles we observe in the hemocytes likely represent a mix of both secretory vesicles and endocytic compartments. In developing *Drosophila*, collagen IV is primarily produced by the fat body, which functions similarly to the liver in mammals (Pastor-Pareja et al, Dev Cell. 2011, 21,245-56). It is secreted and subsequently "captured" at the basal surface of epithelial cells. Hemocytes are believed to endocytose and recycle this collagen IV, and there is evidence suggesting that these cells can also deposit collagen at the basal surface of cells (Bunt et al, Dev Cell. 2010, 19, 296-306). Knocking down collagen IV in an attempt to modulate the number of collagen molecules within the vesicles we report would be far from straightforward, given the complex interplay between collagen production by the fat body, endocytosis-recycling processes, and potentially collagen production by the hemocytes themselves.

Furthermore, the accuracy and reliability of our molecular counting method are supported by the quantitative PAINT (qPAINT) technique, a well-established approach introduced in *Nature Methods* (2016, 13:439–442) by Peng Yin's lab. qPAINT leverages predictable binding kinetics between imager and docking DNA strands to quantify antibody-labeled proteins. Its robustness has been validated on that original publication, including benchmarking against nuclear pore complexes in cellular contexts, with reported accuracy and precision of 95% and 84%, respectively. Thus, while genetic manipulation is not suitable in this case, our approach relies on a rigorously validated counting method with established precision. We also highlight that qPAINT has been utilized in many other contexts, including Cell Reports (2018, 22, 557 – 567); Cell Reports (2020, 33, 108523); Neurophotonics (2019, 6, 035008); Cell Reports Methods (2023, 3, 100408).

We have now included these additional references when citing qPAINT analysis (page 26) to underscore its recognition as a well-established molecular counting method within the DNA-PAINT community.

Point 10: Minor Concerns:

Point 10.(1): On page 2, the authors state, "small field-of-view (FOV) of $\sim 40 \times 10 \mu\text{m}^2$ for HILO excitation". However, even in early demonstrations by the Zhuang group, a FOV of at least $30 \times 30 \mu\text{m}^2$ was achieved for STORM with HILO (PMC2596623). Please clarify and discuss this apparent contradiction, and provide a rationale for why the reported FOV in this study is so limited in comparison.

Response 10.(1): We appreciate the reviewer's comment and the opportunity to clarify this point. The field-of-view (FOV) in HILO excitation can vary significantly depending on multiple experimental factors, including the optical setup, illumination angle, and camera sensor size. We would also like to acknowledge that there was a typographical error in our original statement. The intended FOV was $40 \times 40 \mu\text{m}^2$, not $40 \times 10 \mu\text{m}^2$. We apologize for this mistake, and we have now corrected it in the revised manuscript. Given this correction, our reported FOV is consistent with previous studies, including those by the Zhuang group.

Point 10.(2): On page 6, line 21, "the GFP channel" appears to be a typo. Please correct it to "GFP."

Response 10.(2): Thank you for bringing this to our attention. The typo has been corrected, and "the GFP channel" now reads as "GFP channel." We appreciate your careful review.

Point 10.(3): I find lines 23–27 unclear. What is the relationship between photoreceptor neurons, progenitor cells, pigment cells, accessory cells, the ppm, and the retinal epithelium?

Please clarify this connection and provide additional context to make the relationships between these components more understandable.

Response 10.(3): We thank the reviewer for pointing this out.

We have now simplified this section to make it easier for the reader to focus on the key information.

Page 8, lines 36 – 38: *“As previously reported, E-cadherin is enriched between the newly differentiating photoreceptor neurons and is expressed at lower levels between the remaining progenitor cells that surround the photoreceptors.”*

Point 10.(4): On page 6, line 43, what does “the structural complexity” refer to? How is this concept relevant to epithelial morphogenesis? Please provide a clearer explanation or context to clarify its significance.

Response 10.(4): We have rephrased this sentence to focus it on our specific findings.

Page 9, lines 13 – 21: *“These quantitative measurements provide valuable insights into the spatial organization of E-cadherin at the AJ of a developing epithelium by highlighting that these cell contacts are heterogeneous and present nanoclusters of finite size. This finding is consistent with what has been reported before in the fly embryo using PALM imaging under HILO illumination, with E-cadherin domains ranging from 200 nm to 600 nm in length. Importantly, our new imaging approach extends this previous finding by enabling the examination of E-cadherin organization at greater penetration depths in more complex developing tissues.”*

Point 10.(5): On page 7, line 9, what does the colon “:” between “GFP” and “Collagen-IV” mean?

Response 10.(5): We appreciate the reviewer bringing this to our attention, as there is a typo, the single colon “:” should be double colon “::”. To clarify, the double colon (::) indicates a fusion or combination of two components—in this case, GFP and Collagen-IV. This notation is commonly used in biological and genetic contexts to describe fusion proteins or to report strains. In the context of strains, it specifically denotes that a strain has been engineered to express GFP under the regulation of Collagen-IV promoters or in conjunction with Collagen-IV-related expression. Essentially, the double colon implies a functional or structural relationship between GFP and Collagen-IV in the context of the strain being described.

We have changed the single colon to double colon on page 10, line 5.

Point 10.(6): In Figure 4e, what do the colors represent? Do they indicate Z-depth or intensity differences? Please clarify.

Response 10.(6): The DNA-PAINT image in Figure 4e (as well as all the DNA-PAINT images on this manuscript) has been rendered using the render module of Picasso. In Picasso, each localization is visualized as a Gaussian spot, with the spread determined by either the global or individual localization precision, in our case we opted for the individual localization precision. The rendering module constructs the final image by overlaying Gaussian spots, where each localization contributes one Gaussian intensity distribution. Consequently, the

intensity at each pixel in the image is determined by summing the contributions from all Gaussians overlapping that pixel. Therefore, colors in Figure 4e do not represent Z-depth but instead indicate differences in intensity derived from the summed Gaussian distributions. These intensity variations reflect the density of localizations within the confocal excitation volume.

To clarify this point, we have now added the following section in the Methods section of the paper (page 25, lines 26– 30):

“Image rendering was carried out using the Render module of Picasso. Each localization was represented as a Gaussian spot, with the spread corresponding to the individual localization precision. For each pixel, the intensities of overlapping Gaussian spots were summed to determine the pixel intensity, which was then visualized using an appropriate colormap.”

Point 10.(7): *The title is vague, referencing terms like “sub-10 nm,” “Large Areas,” and “Deep Penetration,” while also including multiple acronyms such as “SDC-OPR” and “DNA-PAINT.” It would be advisable to reconsider the title to make it more precise and aligned with the Nature Communications format.*

Response 10.(7): We appreciate the reviewer’s suggestion regarding the title. We acknowledge the concern about the inclusion of multiple acronyms and have revised the title to enhance clarity and alignment with the *Nature Communications* format.

Regarding the use of “DNA-PAINT” in the title, we note that this acronym has been commonly used in the titles of several *Nature Communications* publications [*Nat Commun* **11**, 4339 (2020), *Nat Commun* **14**, 4345 (2023), *Nat Commun* **12**, 501 (2021), *Nat Commun* **13**, 5047 (2022), *Nat Commun* **10**, 1268 (2019)], demonstrating its recognition and relevance in the field.

Based on the reviewer’s feedback, we propose the following revised title: *“Super-Resolution Imaging in Whole Cells and Tissues via DNA-PAINT on a Spinning Disk Confocal with Optical Photon Reassignment.”*

This revised title improves clarity, reduces the number of acronyms, and conveys the study’s key contributions. We hope this modification addresses the reviewer’s concerns.

Reviewer

3:

*This manuscript demonstrates that DNA-PAINT imaging with a Spinning Disc Confocal and Optical Photon Reassignment (SDC-OPR) enables super-resolution imaging across large fields of view (FOVs) at depths. They benchmarked the approach using DNA-origami constructs, nuclear pore complexes, and microtubules, achieving 5 nm spatial resolution in the basal plane and 10 nm precision at depths of 9 μm within a 53 \times 53 μm^2 FOV. Furthermore, the method’s power and versatility were showcased by imaging the developing *Drosophila* eye epithelium, which revealed distinct E-cadherin populations in adherens junctions with sub-13 nm precision and quantified 46 \pm 27 Collagen IV molecules per vesicle. The relatively high significance of the approach was driven by the super-resolution imaging at large field of view in scattering samples. This manuscript provides such a method and comprehensive quantitative analysis of the resolution at different field of view and depths across various samples.*

Point 1: *However, the manuscript lacks clarity regarding how much the combination of photon reassignment and DNA-PAINT improves resolution and to what extent. The authors do not*

provide a direct comparison between the resolution achieved by SDC with DNA-PAINT vs. SDC-OPR with DNA-PAINT, which makes it difficult to assess the actual benefit of adding photon reassignment. If this improvement is substantial, the authors should discuss its potential applicability to other photon reassignment systems capable of deep imaging with large fields of view, such as instant SIM or multi-view SIM. This could highlight the broader implications for super-resolution microscopy.

Response 1: Thank you for your comment regarding the resolution improvement with the combination of photon reassignment and DNA-PAINT. We understand that the direct comparison between SDC and SDC-OPR with DNA-PAINT is critical for evaluating the actual benefit of adding the photon reassignment module.

We have indeed conducted a benchmark comparison using DNA-origami samples to quantify the improvement in localization precision (and hence resolution) achieved with the SDC-OPR set-up compared to the standard SDC in our original submitted manuscript. This comparison was presented on page 4, lines 6-11:

“As can be seen from the distribution of localization positions, with the SDC, we could not resolve individual docking strands spaced 10 nm apart. More representative DNA-origami images can be found in Supplementary Figure 1a. In contrast, DNA-PAINT on the SDC-OPR (Fig. 1d) successfully resolved individual docking strands, both separated by 17 nm and 10 nm, across all DNA-origami structures within a $53 \times 53 \mu\text{m}^2$ FOV.”

Furthermore, we present the following data in the figure caption of Supplementary Figure 1a to illustrate the difference:

“a) DNA origami structures imaged with SDC at 4x magnification. σ_{SMLM} : 7.0 nm. NeNa value: 11.5 nm. (b) DNA origami structures imaged using 4x magnification with SDC-OPR, with deconvolution as described in methods. σ_{SMLM} : 1.4 nm. NeNa value: 2.3 nm.”

From these values, we observe a marked improvement in the resolution when using the SDC-OPR set-up. Specifically, the NeNa value (the nominal precision of the localization) is reduced from 11.5 nm (with SDC) to 2.3 nm (with SDC-OPR), and the σ_{SMLM} (the standard deviation of the localization precision) improves from 7.0 nm to 1.4 nm. These improvements in localization precision directly correspond to an enhancement in resolution, allowing us to resolve structures that were previously indistinguishable with the SDC alone.

We believe this improvement in resolution is substantial, and as suggested, the enhanced precision could indeed be applicable to other photon reassignment systems capable of deep imaging with large fields of view, such as instant SIM or multi-view SIM. **We have added the following comment on the concluding paragraph in the revised manuscript to highlight how this approach could benefit super-resolution microscopy in a wider context (page 13, lines 16-25).**

“While our study demonstrates the power of photon reassignment microscopy for tissue super-resolution imaging, it remains an open question whether other high-resolution techniques, such as instant structure illumination microscopy (SIM) or multi-view SIM, could achieve similar levels of resolution while enabling even deeper imaging. Given their capacity for structured illumination and multi-angle acquisition, these approaches could, in principle, complement or extend the depth penetration achieved with SDC-OPR while preserving high spatial precision. Future studies comparing these methodologies will be essential to continue pushing the limits of super-resolution microscopy in thick and complex biological samples.”

Point 2: *In addition, it would be beneficial to discuss potential challenges when performing multiplex/multicolor DNA-PAINT with photon reassignment approach.*

Response 2: We thank the reviewer for highlighting the potential challenges of performing multiplex/multicolor DNA-PAINT using the photon reassignment approach. The most widely adopted implementation of multiplex DNA-PAINT imaging is Exchange-PAINT [Nat Methods 11, 313–318 (2014)]. This method employs orthogonal DNA-imager strands conjugated to a single fluorophore, which are all excited using the same laser source. The signals are separated through sequential imaging rounds rather than simultaneous acquisition, making the approach robust and straightforward to implement. Applying Exchange-PAINT to the SDC-OPR system is similarly straightforward, requiring only an imaging chamber with fluid exchange capabilities to enable sequential strand exchanges.

In response to the reviewer's suggestion, we have now demonstrated this approach by performing multiplexed imaging of Nup96-mGFP, mitochondria and microtubules in U2OS cells. Additionally, we have carried out RESI imaging [Nature 617, 711–716 (2023)], which relies on Exchange-PAINT, of Nup96-mGFP in U2OS cells, further showcasing the multiplexing and advanced enhanced resolution capabilities achieve in the SDC-OPR system (i.e., RESI reconstruction achieved an average lateral localization precision of 3 Å, representing a ten-fold improvement over single-round DNA-PAINT imaging). The results of these experiments are included in the new Figure 3 of the revised manuscript.

To address the reviewer's comment on simultaneous multicolor imaging, we have also included a discussion in the revised manuscript (pages 11, lines 15 – 31) regarding the feasibility of simultaneous two-color imaging on commercial SDC-OPR systems. These setups typically incorporate a dual-sCMOS camera configuration, which can enable simultaneous acquisition of two spectrally distinct channels, further expanding the potential for multicolor imaging applications. Specifically, we have added the following paragraph:

“Beyond high-resolution single-color imaging, the SDC-OPR system enables multicolor SMLM experiments, which are essential for studying cellular structures and interactions. In this study, we demonstrate its multicolor imaging capabilities by sequentially imaging Nup96-mGFP, mitochondria and α -tubulin in U2OS cells using orthogonal DNA imager strands, leveraging DNA-PAINT's exchange mechanism for multiplexed imaging. Additionally, we applied RESI imaging, which utilizes Exchange-PAINT, to visualize Nup96-mGFP in U2OS cells with 3 Å localization precision. These results highlight the adaptability of the SDC-OPR system for high-precision multiplexed imaging, further expanding its utility in nanoscale biological investigations. Notably, beyond Exchange-PAINT, multicolor imaging can also be achieved through simultaneous two-color acquisition using the dual-camera configuration commonly available in commercial SDC-OPR systems. This approach can enable real-time acquisition of two spectrally distinct channels, enhancing the potential for multiplexed imaging applications. For such implementations, the recently identified best-performing dyes for 488 nm, 560 nm, and 640 nm excitation could facilitate a combination of sequential and spectral separation strategies within the SDC-OPR configuration, enabling highly multiplexed imaging.”

We believe these additions will be valuable for readers seeking to implement multicolor DNA-PAINT using photon reassignment approaches.

Point 3: *Another limitation of the manuscript is the lack of analysis on axial views and axial resolution. Providing a few examples of how axial resolution is achieved would help clarify its*

effectiveness. The coarse 1 μm axial step size does not provide sufficient information, limiting the ability to assess the system's performance in the axial direction as well as the challenges (like photobleaching and photodamaging) associated with the long imaging acquisition process.

Response 3: We appreciate the reviewer's comment highlighting the importance of analyzing axial views and demonstrating axial resolution in our study. We acknowledge that the coarse 1 μm axial step size initially employed was insufficient for evaluating the system's axial performance. In response, we have refined our approach by performing 3D imaging using the experimental point spread function (PSF) real-time fitter developed by the Ries lab [Nat. Methods, 15, 367–369 (2018)]. This method enables robust 3D localization from 2D data without requiring hardware modifications by leveraging unmodified PSFs to extract accurate z-positions based on the PSF z-size dependence.

Using this method, we demonstrated 3D single-molecule localization microscopy on the SDC-OPR setup by resolving the mitochondrial network in U2OS cells imaged via DNA-PAINT. The results, now included in Figure 5, demonstrate the successful 3D reconstruction of mitochondrial structures, underscoring the feasibility and broader applicability of 3D SMLM imaging on the SDC-OPR system. Specifically, we have added new paragraphs on pages 7 and 8, lines 24-41 and 1-11, respectively, of the revised manuscript, respectively, describing the results. For direct details on these manuscript modifications, we refer the reviewer to our response to Reviewer 1, Point 2.

Regarding concerns about photobleaching and photodamage during extended imaging sessions across different z-planes, we note that, as a confocal-based system, axial excitation is confined to approximately 500 nm per z-step, minimizing photodamage to the sample. We also highlight that to mitigate the risks of photobleaching and photodamage, we employed optimized imaging conditions, including the use of oxygen scavenging buffers, to preserve fluorophore performance over extended acquisition times. These strategies are detailed in the Methods section.

Point 4: *Finally, imaging at a 9 μm depth is not particularly remarkable, especially when using a spinning disk, as the *Drosophila* eye epithelium is relatively transparent. Testing the method on tissues with greater optical scattering and deeper regions would better demonstrate the limits of the approach. Such testing would provide a clearer understanding of how the system performs in more challenging biological contexts.*

Response 4: The *Drosophila* eye imaginal disc is a well-established model epithelium. It is surrounded by a collagen IV-rich basement membrane and lined by hemocytes (macrophage/fibroblast-like cells) and glial cells. The epithelium itself consists of multiple cell types organized in 3D and has a thickness of approximately 20 microns. Given this structure, we are unclear about the reviewer's characterization of the tissue as "relatively transparent." To our knowledge, there is no evidence suggesting that *Drosophila* epithelial cells are more transparent than epithelial cells in other organisms. As such, we do not anticipate major differences in light distortion and scattering when imagining an epithelium like the fly retina compared to a mammalian epithelium, for example.

On the other hand, to address the reviewer's concern about imaging depth, we have now extended our experiments to the pupal fly retina. At this developmental stage, photoreceptors remodel their apical-basal polarity and elongate along the lens-to-brain axis. Importantly, their adherens junctions (AJs) align along this axis, providing a natural testbed to assess the depth

limitations of our imaging approach. We found that we could successfully super-resolve photoreceptor AJs at depths of up to 15 microns—an imaging depth for DNA-PAINT that, to our knowledge, has not been reported with any other imaging setup. These new results, presented in Supplementary Figure 13, are described in page 9, lines 24-33:

“To further assess the capability of DNA-PAINT imaging at greater penetration depths, we extended our imaging to the pupal fly retina, where photoreceptors elongate along the lens-to-brain axis and their AJs align accordingly, providing an opportunity to study imaging performance at increased depth in a tissue context. Unlike the third instar eye disc, which is a thinner, curved epithelial sheet, the pupal retina features a more stratified organization with deeper cellular layers, presenting a more demanding test for high-resolution imaging. Remarkably, we were able to super-resolve photoreceptor AJs at depths of up to 15 microns with a σ_{SMLM} of (26 ± 1) nm and NeNa of (21 ± 1) nm (see Supplementary Figure 13).”

Figure R5 (Supplementary Fig. 13). SDC-OPR DNA-PAINT images of E-cadherin within the *Drosophila* pupal fly retina at (a) 5 μm and (b) 15 μm . Scale bar: 7 μm .

Point 5: Point Minor comments: there are a few unclear statements:

Point 5.(1): Lines 24-26, “line-scanning configurations are, as image formation is realized by progressively scanning single focuses’. Note that line confocal is not a single focus

Response 5.(1): Thank you for your valuable feedback. You are absolutely right—this was a mistake on our part. The phrase should have been “point-scanning” instead of “line-scanning.” We have now rectified this in the revised manuscript to ensure accuracy.

Point 5.(2): Line 38: “However, as the emission light is blocked by the disc, the achievable resolution remained limited.” The emission light passes only through the pinhole, blocking more out-of-focus light to enhance resolution, although the cost of a reduced signal-to-noise ratio.

Response 5.(2): We acknowledge the reviewer’s clarification and agree that the emission light is selectively transmitted through the pinhole rather than completely blocked. The pinhole serves to reject out-of-focus light, thereby enhancing optical sectioning and improving contrast. However, this comes at the cost of reduced photon collection due to the rejection of a significant portion of emitted photons.

We have revised the statement to more accurately reflect this (page 3, lines 6 – 8):

“However, as the emission light passes through the disks, which enhance optical sectioning by rejecting out-of-focus fluorescence, this also reduces photon collection, ultimately constraining the achievable resolution.”

Point 5.(3): Line 13/43: *“This focus contraction..... effectively reducing the effective pinhole size and improving overall photon collection.”* The photo-reassignment process is unrelated, or at least there is no obvious connection, to pinhole size. The claim of ‘effectively reducing the effective pinhole size’ should be revised or clarified to avoid confusion.

Response 5.(3): We appreciate the reviewer’s comment. In the SDC-OPR system, the micro-lens array contracts the emission focus twofold, effectively improving photon collection efficiency. While this contraction does not physically change the pinhole size, it modifies the spatial distribution of detected photons, leading to an effect similar to reducing the effective pinhole size. To avoid confusion, we have revised the text to simply focus on the fact that photon collection increases without the reference to the pinhole size (page 3, lines 21– 22):

“This focus contraction redirects emitted photons to their most probable points of origin, thereby improving overall photon collection.”

Point 5.(4): Page 17, line 18-19, *“.....resulting in a pixel size of 108 nm (with no binning), 78 nm (with 2× binning) and 108 nm (with a 4× binning), respectively”*. It is confusing that both no binning and 4× binning result in a pixel size of 108 nm.

Response 5.(4): The pixel size of 108 nm with both no binning and 4× binning is due to different imaging configurations. When using the 60× objective with 1× magnification, the pixel size is 108 nm, calculated as follows: $6.5 \mu\text{m}$ camera pixel size \div 60 = $0.108 \mu\text{m}$ (108 nm). Adding 4× magnification (increasing the effective magnification by a factor of 4) reduces the pixel size to 27 nm, calculated as: $6.5 \mu\text{m} \div (60 \times 4) = 0.027 \mu\text{m}$ (27 nm). To achieve a suitable pixel size for single-molecule localization microscopy, digital binning is applied. With 4× binning, 16 camera pixels are grouped together, effectively returning the pixel size to 108 nm ($27 \text{ nm} \times 4 = 108 \text{ nm}$). This is why the original sentence reads: “Magnification on the SDC-OPR can be 1×, 2.8×, and 4×, resulting in a pixel size of 108 nm (with no binning), 78 nm (with 2× binning), and 108 nm (with 4× binning), respectively.”

To provide further clarity, we have now modified this sentence to read:

“Magnification on the SDC-OPR can be 1×, 2.8× and 4×, resulting in a pixel size of 108 nm (with no binning, 1× magnification), 78 nm (with 2× binning, 2.8× magnification) and 108 nm (with a 4× binning, 4× magnification).”

Point 5.(5): Page 19, line 9, *“.....using 17500 frames and an integration time of 300 ms frame”*. Is the frame # for all slices or single slice? What is the total acquisition time for all depths?

Response 5.(5): The number of frames corresponds to each z-step, with an integration time per frame of 300 ms. The total acquisition time depends on the size of the cell and the number of z-steps acquired. For a $9 \mu\text{m}$ cell with a $1 \mu\text{m}$ z-step, the total acquisition time with 17,500 frames per z-step is 13.1 hours. However, depending on the structure of interest, the number of frames can be reduced. For example, we show here (Figure R1, for review only) that for

microtubules, only 2500 frames are sufficient to reconstruct their distribution in cells, which would reduce the acquisition time from 13.1 hours to less than 2 hours for the whole cell.

a) $z = 0 \mu\text{m}$

b) $z = 4 \mu\text{m}$

Figure R6. Example resulting DNA-PAINT images of microtubules in HeLa cell when filtered for shorter frame acquisition times. (a) Microtubules at basal plane of HeLa cell (penetration depth of $0 \mu\text{m}$) showing resulting image when filtered for different frame length acquisitions (2500, 5000, 7500, 10000, 12500 and 15000 frames). (b) Microtubules at higher penetration depth of HeLa cell ($4 \mu\text{m}$), again filtered for the same frame length acquisition times as in (a). Cells were stained with DNA-labelled anti-alpha tubulin antibodies, as described in the Methods section. Scale bar is $6 \mu\text{m}$.

Additionally, we have added a section in the Discussion addressing the implementation of different types of fluorogenic imager strands (dye-quencher or dye-dye-based self-quenching

imager probe) to increase imager concentration without loss of resolution, thereby improving imaging speed (pages 12 and 13, lines 34 – 41 and 1-5, respectively):

“Building on this foundation, we anticipate that the ability to achieve such high-resolution imaging at depth will open new avenues for investigating a wide range of biological questions, including the dynamics of cellular adhesion, tissue morphogenesis, and extracellular matrix organization in various developmental contexts. Recent advancements in DNA-PAINT imaging have introduced dye-quencher or dye-dye-based self-quenching imager probes, which significantly reduce background fluorescence, enhance brightness, and improve spatial resolution while achieving faster imaging speeds. Incorporating these fluorogenic probes within the SDC-OPR system could further amplify imaging speed and resolution. By marrying these emerging innovations with the capabilities of the SDC-OPR platform, future research could address even more complex biological systems, making this approach a versatile tool for studying cellular and molecular architecture across diverse tissue types.”

Point 5.(6): Line 36, “Drift correction was applied using redundant cross-correlation, utilizing gold particles as fiducials for cellular experiments.” The author may want to provide more details on drift correction (both lateral and axial shifts?), the size and localization accuracy of gold particles, how many are needed for large FOVs, how they are distributed, and the extent of lateral and axial resolution correction.

Response 5.(6): Thank you for your helpful suggestion. The size of the gold nanoparticles used in our experiments is 90 nm, and we have already provided details regarding their size and incubation time in the Methods section of the originally submitted manuscript: *“The sample was washed again with buffer B+ and 100 μ l of gold nanoparticles (90 nm, no. G-90-100, Cytodiagnosics) was flushed through and incubated for 5 min before washing with buffer B+.”*

Gold nanoparticles are commonly used as fiducials in the SMLM community, especially when employing DNA-PAINT with a 561 nm excitation laser. Under these experimental conditions, we typically observe 50 gold nanoparticles within the field of view (FOV), which are localized with an accuracy of 1 nm. This provides reliable drift correction, particularly for lateral shifts at the focal plane ($z = 0$). However, as the imaging moves to higher z -planes, these fiducials are no longer detectable. In such cases, we apply a drift correction method based on the adaptive intersection maximization (AIM) technique, developed by Hongqiang Ma et al. [Sci. Adv.10, eadm7765 (2024)], which is implemented in the Picasso platform.

To clarify further, we have now expanded the explanation of the DNA-PAINT analysis and the drift correction methods in the Methods section of the manuscript as follows (page 25, lines 19 – 25):

“Typically, around 50 nanoparticles were detected within the field of view (FOV), providing reliable drift correction, particularly for lateral shifts at the focal plane ($z = 0$). At higher z -planes, however, gold fiducials were no longer detected. In such cases, drift correction was performed using the adaptive intersection maximization-based method (AIM), a marker-free algorithm developed by Hongqiang Ma et al., which is implemented in the Picasso platform.”

Response to Reviewer Comments
Nature Communications ID NCOMMS-24-69494A

Reviewer's comments in *italic*

Author's responses in blue. Changes to the Manuscript in red.

Reviewer 1

The authors of the manuscript entitled "Super-Resolution Imaging in Whole Cells and Tissues via DNA-PAINT on a Spinning Disk Confocal with Optical Photon Reassignment" have addressed all concerns I raised during the previous round of revision. They have included new experimental results demonstrating the sub-10 nm resolution imaging capabilities of the SDC-OPR system, multiplexed Exchange PAINT imaging and RESI workflow, and 3D imaging results. All these new findings improve the manuscript substantially. It also strengthens and broadens the capabilities of the SDC-OPR workflow. Therefore, I would recommend proceeding to the publication of this manuscript in Nature Communications.

Response: We sincerely thank Reviewer 1 for their thoughtful review and positive feedback. We appreciate the constructive comments that helped improve our work and are glad the revisions have addressed all concerns.

Reviewer 2

First of all, I would like to express my appreciation for the efforts by Zaza et al. to address many of the points raised in my previous review, including multi-color imaging, 3D imaging, mechanistic insights into the improvements enabled by SDC-OPR, uneven illumination correction, and demonstration of applications in cultured cells. Most of these issues have been adequately addressed, and the revised manuscript is now closer to being suitable for publication in Nature Communications.

Response: We thank Reviewer 2 for the sharp reading and positive appraisal of our revised manuscript, as well as for raising comments that had allowed us to enrich the work.

However, one important issue previously raised remains insufficiently addressed: the application of this method to tissue in *Drosophila*, which pertains to Points 6 and 9 in my prior critique. I hope the authors can provide adequate responses to the following questions.

Major Points:

Point 1: I appreciate the clarification in the revised manuscript regarding E-Cadherin distribution. The observed high-intensity foci between cells, which vary in size, are consistent with adherens junctions (AJs) being punctate or patchy, as seen in cell types such as neuroblasts or migrating cells. While this may be the first demonstration in which such size heterogeneity is measured using super-resolution microscopy, I am unsure whether the heterogeneous distribution of E-Cadherin has truly never been reported before. Therefore, the following sentence on page 8, line 31, may be overstated and should be moderated: "This heterogeneity, previously unreported in *Drosophila* retinal development..."

Response 1: We appreciate the reviewer's comment and agree that the statement should be moderated to reflect prior observations of E-Cadherin heterogeneity. In the revised manuscript, we have revised the wording to acknowledge previous reports while emphasizing that this is the first time such heterogeneity has been observed at this level of resolution in *Drosophila* retinal development and at this penetration depth (page 8).

"This represents the first demonstration of size heterogeneity at nanometer resolution in Drosophila retinal development, echoing similar findings in the developing embryonic epidermis".

Point 2: E-Cadherin is typically localized at adherens junctions, particularly at the zonula adherens (ZA) on the apical side of epithelial cells. However, in the eye, its distribution appears to extend along the apical-to-basal axis, as shown in Figure 6c. Is this broader distribution previously known, or does it represent a novel observation? If the authors indeed observe E-Cadherin localization spanning this axis, do they detect any differences in the pattern or degree of heterogeneity between the apical and basal regions? Addressing this question could provide further insights into the spatial regulation of E-Cadherin and its role in retinal development.

Response 2: We appreciate the reviewer's insightful question. To clarify, the E-cadherin distributions shown in Figure 6c correspond to two distinct epithelia: the squamous peripodial membrane (ppm) epithelium ($z = 0 \mu\text{m}$) and the underlying columnar retinal epithelium ($z > 0 \mu\text{m}$), as depicted in Figure 6a.

Within the retinal epithelium, E-cadherin is localized at the ZA of the retinal progenitor cells ($z = 4 \mu\text{m}$, $z = 5 \mu\text{m}$; Fig. 6c, Fig. 6d) and at the ZA of the photoreceptor neurons, which are found slightly below the plane of the progenitor cells (at $z = 9 \mu\text{m}$), at the apical pole of cells. E-cadherin is not observed below the ZA, when imaging the lateral-basal membrane. Therefore, the distribution in Figure 6c does not represent an apical-to-basal extension within a single epithelium but rather reflects distinct ZA domains belonging to two epithelia.

While we have clarified this in the revised manuscript, we've also extended our analysis by investigating potential differences in E-cadherin size distribution between the ppm epithelium ($z = 0 \mu\text{m}$) and the retinal epithelium ($z = 5 \mu\text{m}$). To explore this, we performed the same clustering analysis of E-cadherin foci in both tissues. Interestingly, despite these two epithelia cell types being different – squamous *versus* columnar– our analysis revealed that E-cadherin exhibits a similar minimal foci size in both epithelia. This suggests that the minimal E-cadherin foci may represent a fundamental property of E-cadherin-mediated adhesion, rather than a tissue-specific characteristic.

We have now included this quantification in Figure 6d and expanded on this point in the manuscript with the following changes:

"To further investigate the size distribution of E-cadherin foci in both epithelial layers, we applied density-based clustering analysis (DBSCAN) to DNA-PAINT data at $z = 0 \mu\text{m}$ (squamous peripodial cells) and $z = 5 \mu\text{m}$ (progenitor columnar cells of the eye disc). This analysis identified a minimum E-cadherin foci size of $(230 \pm 40) \text{ nm}$ across both cell types. However, columnar cells exhibited larger foci domains, reaching $(352 \pm 90) \text{ nm}$ and $(470 \pm 160) \text{ nm}$, corresponding to 1.5x and 2x the size of the smallest foci, respectively (Fig. 6d) (see Supplementary Fig. 12 for line profile comparisons of GFP vs DNA-PAINT images at $5 \mu\text{m}$ depth). These quantitative measurements provide valuable insights into the spatial

organization of E-cadherin at the AJ of a developing epithelium by highlighting that these cell contacts are heterogeneous and present nanoclusters of finite size. This finding is consistent with what has been reported before in the fly embryo using PALM imaging under HILO illumination, with E-cadherin domains ranging from 200 nm to 600 nm in length. Notably, our study extends these previous findings by demonstrating that this nanometer-scale organization is conserved across distinct epithelial tissues, suggesting that the minimal E-cadherin foci size may represent a fundamental property of E-cadherin-mediated adhesion. Importantly, our new imaging approach extends this previous finding by enabling the examination of E-cadherin organization at greater penetration depths in more complex developing tissues.”

Here is the expanded version of Figure 6, with the updated panel 6d:

Figure 6. (d) Zoomed-in view of the highlighted region from (c), showing DBSCAN clustering analysis of the E-cadherin DNA-PAIN image. Histogram displays the maximum distances within clusters, derived from the sample shown in (c) for z = 0 μm and z = 5 μm .

Point 3: Point 9 from my previous review remains critical. Without appropriate control experiments, it is difficult to determine whether the reported quantitative measurements in tissue are definitive. While qPAINT may offer reliable quantification, my concern lies in the feasibility of performing such quantification in complex tissue environments. The authors should consider using target proteins for which genetic or pharmacological manipulations are available to validate quantification in tissue samples.

Response 3: We appreciate the reviewer’s concerns regarding the feasibility of performing quantitative qPAINT measurements in complex tissue environments. To address this, we emphasize that qPAINT quantification in tissue follows the same fundamental principles as in single-cell or in vitro contexts. qPAINT relies on analyzing the binding kinetics between imager and docking strands, specifically by measuring the average dark time (i.e., the waiting time between binding events) of a cluster of single-molecule localizations. The inverse of the measured dark time, known as the influx rate (ξ) or qPAINT index, is directly proportional to the number of proteins within that cluster, enabling robust molecular quantification. Crucially, we observe continuous binding and unbinding events of the DNA imager strands in the tissue context, as expected, confirming that molecular counting in tissue environments functions identically to well-established cellular qPAINT applications. We have now presented examples of such single-molecule traces in Supplementary Figure 15.

Furthermore, qPAINT quantification approach relies on internal calibration within the same sample, leveraging the detection of single target molecules within the field of view. This ensures that molecular counts remain unaffected by potential variations in probe accessibility or labelling efficiency. The robustness of this approach is further supported by independent replicate experiments, as presented now in Supplementary Figure 15b, demonstrating that the measured molecular numbers are highly reproducible across different tissue samples.

While additional validation using genetic or pharmacological manipulations could further strengthen qPAINT's applicability in tissue, our study provides strong evidence that qPAINT-based molecular quantification is reliable in complex tissue environments. Moreover, we note that even in the original qPAINT publication (*Nat Methods* **13**, 439–442 (2016)), the authors demonstrated the feasibility of performing qPAINT in a tissue setting—specifically in the *Drosophila* neuromuscular junction—though under TIRF illumination. By successfully applying qPAINT at greater penetration depths than this previously established example, our work significantly expands the applicability of qPAINT and establishes its potential as a powerful tool for quantitative super-resolution imaging in biological tissues.

To improve the description of qPAINT methodology and calibration experiments, we have now expanded that section to include the following:

“Using a strain where Collagen-IV is endogenously labelled with GFP (GFP::Collagen-IV), we were able to visualize intracellular vesicles containing Collagen-IV in hemocytes localized at the basal surface of the developing retina. To quantify the number of Collagen-IV molecules within these vesicles, we employed quantitative DNA-PAINT (q-PAINT) analysis. qPAINT relies on the analysis of binding kinetics between imager and docking strands, specifically by measuring the average dark time (i.e., the waiting time between binding events) of a cluster of single-molecule localizations, as depicted in Supplementary Fig. 15a. The inverse of the measured dark time, known as the influx rate (ξ) or qPAINT index, is directly proportional to the number of proteins within that cluster, enabling robust molecular quantification.

Using qPAINT analysis, we determined that each vesicle contained an average of 46 ± 27 Collagen-IV molecules (Fig. 6e). As previously reported, Collagen-IV vesicles measure approximately 300 nm in all dimensions, and imaging a single focal plane was sufficient to capture their full axial extent. Importantly, qPAINT-based molecular quantification relies on internal calibration within the same sample, leveraging the detection of single Collagen-IV molecules within the field of view. This approach has proven to be highly reproducible across independent experiments, as demonstrated by two independent replicates presented in Supplementary Fig. 15b, further underscoring the robustness of this method for precise molecular quantification in a tissue context (Supplementary Fig. 15).”

Furthermore, we expanded Supplementary Figure 15 to illustrate the qPAINT methodology, provide examples of two independent replicate experiments for calibration and quantification, and showcase single-molecule traces.

Supplementary Figure 15: Quantitative-Paint (qPAINT) calibration. (a) In DNA-PAINT, fluorescently tagged imager strands transiently bind to complementary docking strands anchored to a target—in this case, GFP-Collagen-IV proteins within vesicles. The fluorescence intensity traces generated from each vesicle's localization clusters over time exhibit distinct on- and off-time patterns, enabling the estimation of the number of molecules within a non-resolvable protein cluster—in this case a collagen vesicle. By determining the binding frequency of a single protein, the number of proteins in different clusters can be estimated by calculating the ratio of their binding frequencies. (b) Left: DNA-PAINT Collagen-IV imaging of the basement membrane for two independent field-of-views (FOVs). Right: Histogram of qPAINT indexes (ξ) for GFP:Collagen-IV proteins for each FOV. The fit to a sum

of two Gaussian functions is shown as solid lines, whereas each component is shown as a dashed line. This fitted value ξ was then used to calculate the number of proteins per cluster in the image. Inset shows blinking kinetics of an example single protein cluster. (c) Zoomed-in views of two example vesicles from FOV1, containing 65 and 20 proteins, respectively, along with their corresponding time traces, where the different binding frequency can be seen. (d) Histogram of proteins per vesicles for FOV1. (e) Zoomed-in views of two example vesicles from FOV2, containing 48 and 5 proteins, respectively, along with their corresponding time traces, where the different binding frequency can be seen. (f) Histogram of proteins per vesicles for FOV2.

Point 4: Additionally, what is the biological significance of measuring collagen molecule numbers with such precision? What physiological insights can be gained from this? As it stands, this point is unclear, and I do not find it particularly compelling in its current form.

Response 4: We appreciate the reviewer's comment and the opportunity to clarify the goal of our experiment. The primary objective of our study was not to reveal new biological insights regarding collagen's role in development, disease, or tissue repair. Rather, our aim was to demonstrate the capability of qPAINT for imaging and quantifying molecular distributions in tissue contexts, including at greater penetration depths than previously reported. Collagen was used as a proxy cargo to illustrate the ability of qPAINT to measure the amount of cargo within trafficking vesicles, serving as a proof of principle for applying qPAINT to more complex biological samples. This study is focused on showcasing the technical capabilities of qPAINT, and while the biological relevance of collagen quantification could be explored in future work, the emphasis here is on demonstrating the potential of qPAINT as a tool for high-resolution, quantitative imaging in tissue.

Minor Points:

Point 5a: • Page 8: Figure references are incorrect. "Fig. 64c" and "Fig. 64d" should be corrected to "Fig. 6c" and "Fig. 6d."

Response 5a: We appreciate the reviewer for pointing this out. The incorrect figure references have been corrected in the revised manuscript.

Point 5b: • Figure citations are inconsistent throughout the manuscript, alternating between "(Figure xx)" and "(Fig. xx)." Please standardize these in accordance with Nature Communications formatting guidelines.

Response 5b: Thank you for pointing this out. We have standardized all figure citations in accordance with *Nature Communications* formatting guidelines.

Point 5c: • Supplementary Figure 12(c): Are the lines drawn along cell-cell contact sites? The current lines obscure the underlying structures, making it difficult to determine the precise cellular positions. Consider making the lines more transparent to allow better visualization of the underlying features.

Response 5c: We appreciate this suggestion. The contrast and lines in Supplementary Figure 12(c) have been changed for dashed boxes to ensure the visualization of the plot structure.

Also zoom files were rotated 180 degrees. We realized that there was a rotation between figures (a) and (b) and the zoom views.

Reviewer #3 (Remarks to the Author):

The authors have addressed my comments, and the manuscript's clarity and quality have been significantly improved.

Response: We sincerely thank Reviewer 3 for their thoughtful review and appreciation of our revisions. We are glad the improvements have enhanced the clarity and quality of the manuscript.